# Random Is All You Need: Random Noise Injection on Feature Statistics for Generalizable Deep Image Denoising

**Zhengwei Yin, Hongjun Wang, Guixu Lin, Weihang Ran, Yinqiang Zheng**[*]
Department of Mechano-Informatics, The University of Tokyo
`zhengwei.yin.default@gmail.com yqzheng@ai.u-tokyo.ac.jp`
`{hjwang-ai,linguixu831,ran-weihang}@g.ecc.u-tokyo.ac.jp`

## Abstract

Recent advancements in generalizable deep image denoising have catalyzed the development of robust noise-handling models. The current state-of-the-art, Masked Training (MT), constructs a masked SwinIR model which is trained exclusively on Gaussian noise ($\sigma$=15) but can achieve commendable denoising performance across various noise types (*i.e.* speckle noise, poisson noise). However, this method, while focusing on content reconstruction, often produces over-smoothed images and poses challenges in mask ratio optimization, complicating its integration with other methodologies. In response, this paper introduces RNINet, a novel architecture built on a streamlined encoder-decoder framework to enhance both efficiency and overall performance. Initially, we train a pure RNINet (only simple encoder-decoder) on individual noise types, observing that feature statistics such as mean and variance shift in response to different noise conditions. Leveraging these insights, we incorporate a noise injection block that injects random noise into feature statistics within our framework, significantly improving generalization across unseen noise types. Our framework not only simplifies the architectural complexity found in MT but also delivers superior performance. Comprehensive experimental evaluations demonstrate that our method outperforms MT in various unseen noise conditions in terms of denoising effectiveness and computational efficiency (lower MACs and GPU memory usage), achieving up to 10 times faster inference speeds and underscoring it's capability for large scale deployments.

## 1 Introduction

Image denoising is a critical area of research in low-level image processing aimed at recovering clean images from noisy counterparts. The rapid advancements in deep learning have inspired numerous studies proposing specialized image denoising networks. These networks, typically trained on pre-defined noise distributions, show remarkable performance in noise removal. However, their generalization to other noise types is limited, restricting their application in real-world scenarios where noise distributions often vary from those in the training phase.

In the current research trend on the image denoising task, most existing works (*i.e.* SwinIR (Liang et al., 2021), Restormer (Zamir et al., 2022)) train and evaluate models on images corrupted with Gaussian noise, which limits their performance to this specific noise distribution. To address this limitation, some methods (Zhang et al., 2017) assume an unknown noise level for a particular noise type, while others (Brooks et al., 2019b; Wei et al., 2020) attempt to improve the performance in real-world scenarios by synthesizing or collecting training data closer to the target noise or directly performing unsupervised training on the target noise (Chen et al., 2018; Yuan et al., 2018). Despite these efforts, recent work by Chen et al. (Chen et al., 2023) argues that none of these methods substantially improve the generalization performance of denoising networks, and they still struggle when the noise distribution is mismatched (Abdelhamed et al., 2018b). In response, they propose masked training and construct a masked SwinIR model which learns the reconstruction of image textures and structures

---

[*]Corresponding author

rather than overfitting to a specific noise type, their model is trained on Gaussian noise $\sigma = 15$ but can generalize well to other different unseen noise types. Nevertheless, we notice that despite the enhanced performances, their model also introduces unwanted side-effects that tends to over-smooth image contents, leading to the loss of high-frequency details and a drop in PSNR (refer to Fig. 1). The generalization challenge in deep denoising continues to be a significant hurdle for broader applicability.

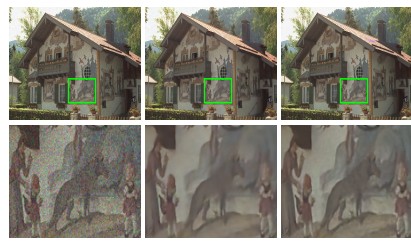

Noisy Image    MT (29.87 dB)    Ours (31.67 dB)

Figure 1: The side-effects of MT (Chen et al., 2023) on image quality: Over-smoothing of content results in a decrease in PSNR. In contrast, our method preserves more details while removing noise, thereby achieving higher PSNR.

In this paper, we present RNINet, a novel architecture built on a streamlined encoder-decoder framework to enhance both efficiency and overall performance for generalizable deep image denoising. Initially, we train a pure RNINet (only simple encoder-decoder) on individual noise types and observe that feature statistics, such as mean and variance, shift in response to different noise conditions (refer to Fig. 2). Some recent studies (Liu et al., 2021; 2023; Chen et al., 2023) have conduct generalization analysis experiment based on feature statistics distribution, but none of them conduct manipulations directly on the learned feature statistics. Leveraging these insights, we incorporate noise injection blocks within RNINet to inject random noise on feature statistics, thereby creating noised features that influence the model's learning. While feature statistics can contain domain-specific information (Huang & Belongie, 2017; Li et al., 2021), this noise injection manipulation allows the noised feature statistics to represent potential unseen noise domains, significantly enhancing the model's generalization capabilities. The main contributions of this work are summarized as follows:

- We present RNINet, a novel architecture that utilizes a streamlined encoder-decoder framework to both enhance efficiency and improve the performance of generalizable deep image denoising. This approach simplifies the architectural complexity typically found in existing generalizable denoising methods, facilitating broader application to real-world deployment environments.

- We introduce a noise injection block that injects random noise into feature statistics, aimed at potential unseen noise domains. This development significantly enhances generalization capabilities, distinguishing our approach from existing research that primarily focuses on generalization analysis without direct operational interventions.

- Our pipeline is straightforward yet highly effective. Comprehensive experiments demonstrate that RNINet surpasses the performance of the state-of-the-art method MT in various unseen noise conditions, delivering superior denoising effectiveness and computational efficiency (lower MACs and GPU memory usage), achieving up to 10 times faster inference speeds and underscoring it's capability for large scale deployments.

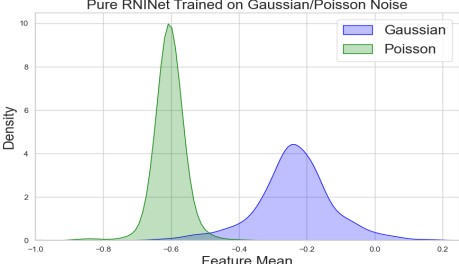
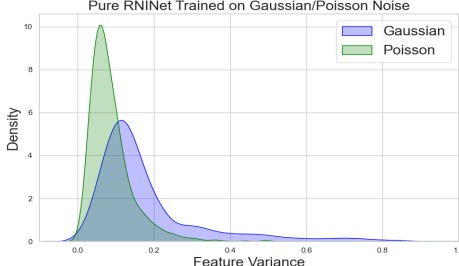

Figure 2: Impact of training with different noise on feature statistics. We investigate the shifts in mean and variance by initially training a pure RNINet (only simple encoder-decoder) with Gaussian and Poisson noise. The feature statistics exhibit markedly different distributions for each noise type.

## 2 RELATED WORK

### 2.1 IMAGE DENOISING

Image denoising techniques predominantly fall into two categories: traditional model-based methods and data-driven deep learning approaches. Traditional methods (Buades et al., 2005; Dabov et al., 2007; Elad & Aharon, 2006; Gu et al., 2014; Mairal et al., 2009) generally rely on modeling image priors to recover content from images affected by noise, demonstrating a certain level of flexibility and generalization capability (Abdelhamed et al., 2018b) across various noise types. However, these methods often struggle to reconstruct fine image details and achieve high PSNR. In contrast, data-driven deep learning models have achieved remarkable denoising performance. CNN models (Zhang et al., 2017; 2022; Lefkimmiatis, 2018; 2017; Mao et al., 2016; Divakar & Venkatesh Babu, 2017; Jia et al., 2019; Zhang et al., 2018) were once the mainstream in denoising models, offering substantial performance improvements over traditional methods. The advent of Vision Transformers (Dosovitskiy et al., 2020), which treat pixels as tokens and leverage self-attention to parse token interactions, has marked a significant paradigm shift. Variants based on Vision Transformers (Zamir et al., 2022; Liang et al., 2021; Zhao et al., 2023; Zhang et al., 2023a; Wang et al., 2022; Chen et al., 2021; 2022b; Yin et al., 2024a;b) have largely supplanted CNN models as the mainstream solution due to their enhanced capability to capture global dependencies. Despite these advancements, a prevalent issue is the training of models on noise patterns identical to those encountered during testing, where the primary performance metric becomes the network's capacity to overfit training noise.

### 2.2 GENERALIZATION PROBLEM

The generalization dilemma in low-level vision tasks, such as image denoising, often emerges when there is a discrepancy between training and testing degradations. Conventionally, models are trained on Gaussian noise, a practice misaligned with the predominantly non-Gaussian noise encountered in real-world scenarios, leading to performance degradation. To address this, solutions have diverged into two primary methodologies: one seeks to simulate real-world noise more closely during training (Brooks et al., 2019b; Wei et al., 2020; Chen et al., 2018; Guo et al., 2019; Plotz & Roth, 2017; Krull et al., 2019; Abdelhamed et al., 2018b), while the other develops 'blind' denoising models assuming that the noise level is unknown or training on a large amount of noise types (Krull et al., 2019; Yue et al., 2019; Zhang et al., 2023b; 2017; Ji et al., 2023b;a; 2024). Recent work by Chen (Chen et al., 2023) has pointed out that these efforts do not sufficiently delve into the generalization shortfalls; these methods still fail to generalize to noise types not represented in the training dataset. Some studies (Liu et al., 2021; 2023) have attempted to analyze the reasons behind poor generalization capabilities of super-resolution models and have identified that conventional training methods tend to make models overfit specific degradation type to achieve higher PSNR. Building on these insights, Chen et al. (Chen et al., 2023) introduced masked training and constructed a masked SwinIR (Liang et al., 2021), designed to focus on content reconstruction rather than overfitting specific noise types. While their approach achieved commendable results, it also introduced some unwanted side-effects: the image content tends to be over-smoothed, leading to a loss of high-frequency details and a drop in PSNR performance. Most recently, Cheng (Cheng et al., 2024) enhanced generalization performance by incorporating a pretrained CLIP model. However, because the CLIP model has been exposed to prior information from billions of images containing diverse noise types, their method is categorized into another track and often excluded from direct comparisons with benchmark methods that were solely trained with Gaussian noise, to ensure fairness.

### 2.3 FEATURE STATISTICS IN NEURAL NETWORKS

Feature statistics (*i.e.* mean and variance) are commonly utilized in the analysis of various neural networks, with research (Huang & Belongie, 2017; Li et al., 2021) indicating that they can capture informative characteristics of specific domains (e.g., color, texture, and contrast). In scenarios involving out-of-distribution data, feature statistics often demonstrate inconsistencies with those of the training domain due to differing domain characteristics (Wang et al., 2019; Gao et al., 2021), while some normalization methods (Ioffe & Szegedy, 2015; Li et al., 2022) can also improve model performance by manipulating feature statistics. Recent studies in the field of low-level vision have also explored model generalization capabilities with feature statistics. For instance, (Liu et al., 2023) proposed a metric based on feature statistics to assess the generalization ability of

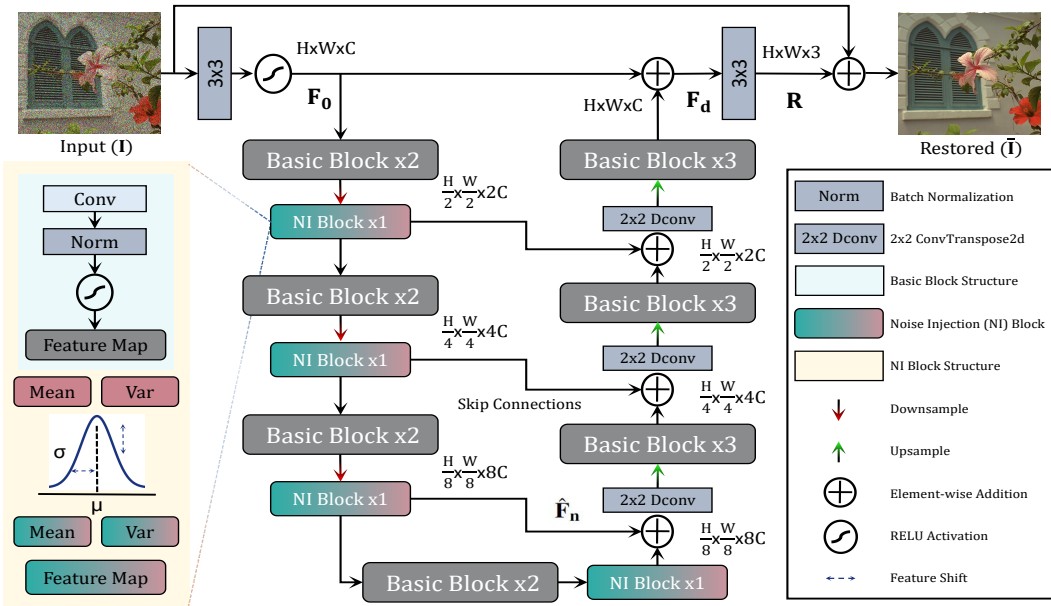

Figure 3: **Overview of RNINet.** Our model consists of a streamlined hierarchical encoder-decoder architecture that integrates basic blocks and noise injection blocks. The basic blocks function as feature extractors within both encoder and decoder stages. Noise injection blocks enhance generalization capabilities by injecting random noise into the feature statistics in the encoder stage.

super-resolution models, while (Liu et al., 2021) introduced the CHI evaluation score following dimensionality reduction and clustering based on feature statistics. Subsequently, MT (Chen et al., 2023) utilized the metric from (Liu et al., 2021) to validate the generalization performance of their generalizable denoising models. However, these studies typically treat feature statistics as deterministic values obtained from the features and rely solely on statistical analysis of these values to validate the generalization capability and efficacy of their methods. In contrast, our approach provide a novel perspective that injects random noise tensors to alter feature statistics, thereby enhancing the generalization capability of our denoising models.

# 3 METHOD

## 3.1 OVERALL PIPELINE

The overall structure of RNINet is depicted in Fig. 3. In the general inference manifold, given a noisy input image $\mathbf{I} \in \mathbb{R}^{H \times W \times 3}$, our proposed RNINet commences by extracting low-level features $\mathbf{F_0} \in \mathbb{R}^{H \times W \times C}$ through a convolution operation followed by a ReLU activation function, where $H \times W$ represents the spatial resolution, and $C$ denotes the number of channels. Subsequently, these feature embeddings $\mathbf{F_0}$ are processed via a four-level hierarchical encoder-decoder structure to transform into deep features $\mathbf{F_d} \in \mathbb{R}^{H \times W \times C}$. Each encoder-decoder level incorporates multiple basic blocks, each consisting of a convolution layer, a batch normalization layer, and a ReLU activation layer. The encoder progressively reduces spatial resolution while enhancing channel capacity, culminating in a low-resolution latent representation $\hat{\mathbf{F}}_n$. To facilitate the encoding process, noise injection blocks are strategically integrated at every two levels of the encoder. These blocks obtain noised feature statistics $\tilde{\mathbf{F}}_n$ by injecting random noise tensors. The decoder's objective is to incrementally reconstruct the high-resolution clean output from $\hat{\mathbf{F}}_n$. Downsampling and upsampling within the features are executed using convolution and transposed convolution, respectively. The refined deep features $\mathbf{F_d}$ are subsequently processed through a final convolution layer to produce a residual image $\mathbf{R} \in \mathbb{R}^{H \times W \times 3}$, which is added to the degraded input to yield the restored image: $\bar{\mathbf{I}} = \mathbf{I} + \mathbf{R}$. Following this, we detail the specific modules comprising the basic and noise injection blocks.

## 3.2 BASIC BLOCK

To mitigate the risk of overfitting and thereby enhance the generalization capabilities of our denoising model, as indicated by existing studies (Liu et al., 2023; Chen et al., 2023; Liu et al., 2021), we have opted for a straightforward structure for the basic block within our RNINet framework. As depicted in Fig. 3, the basic block consists of three layers and functions as feature extractor at both the encoder and decoder stages of the network. Given the input features $\mathbf{F} \in \mathbb{R}^{B \times H \times W \times C}$, the transformation process facilitated by the basic block is defined by the following equation:

$$\mathbf{F_e} = \text{ReLU}(\text{BN}(\text{Conv}(\mathbf{F}))) \tag{1}$$

where Conv denotes the convolution operation, BN represents batch normalization, and ReLU is the rectified linear unit activation function. This streamlined structure ensures efficient and effective feature extraction, with reduced complexity to prevent overfitting issue.

## 3.3 NOISE INJECTION BLOCK

The noise injection block is designed to generate noised features with altered statistical properties, enhancing the model's generalization to unseen noise types. Given feature $\mathbf{F_e}$, this block first conducts downsampling via convolution (excluding the final block), after which the features undergo batch normalization and ReLU activation, resulting in $\hat{\mathbf{F}}_\mathbf{e} \in \mathbb{R}^{B \times H^\mathbf{s} \times W^\mathbf{s} \times C^\mathbf{s}}$, where $H^\mathbf{s}, W^\mathbf{s}, C^\mathbf{s}$ denote the new height, width, and channel dimension after downsampling. The downsampling step increases the channel dimension, aiding in the computation of feature statistics in the subsequent process. Random noises are then injected into these statistics to generate features with altered characteristics.

### 3.3.1 FEATURE STATISTICS

Building on prior research (Li et al., 2021; Huang & Belongie, 2017; Li et al., 2022), we recognize that feature statistics, specifically the mean and standard deviation, retain informative properties of a domain. We compute these statistics for the downsampled feature representation $\hat{\mathbf{F}}_\mathbf{e}$, where $\mu \in \mathbb{R}^{B \times C^\mathbf{s}}$ and $\sigma \in \mathbb{R}^{B \times C^\mathbf{s}}$ represent the channel-wise mean and standard deviation of each instance in a batch, respectively:

$$\mu = \frac{1}{H^\mathbf{s}W^\mathbf{s}} \sum_{h=1}^{H^\mathbf{s}} \sum_{w=1}^{W^\mathbf{s}} \hat{\mathbf{F}}_\mathbf{e} \tag{2}$$

$$\sigma^2 = \frac{1}{H^\mathbf{s}W^\mathbf{s}} \sum_{h=1}^{H^\mathbf{s}} \sum_{w=1}^{W^\mathbf{s}} (\hat{\mathbf{F}}_\mathbf{e} - \mu)^2 \tag{3}$$

Before injecting noise, we standardize the features using Z-score normalization. This step is crucial as it prepares the baseline features by aligning the distribution around a zero mean and unit variance, ensuring the consistency of baseline features in subsequent manipulations. The normalized features, denoted as $\hat{\mathbf{F}}_\mathbf{z}$, are calculated as follows:

$$\hat{\mathbf{F}}_\mathbf{z} = (\hat{\mathbf{F}}_\mathbf{e} - \mu)/\sigma \tag{4}$$

### 3.3.2 REPARAMETERIZATION AS NOISE INJECTION

Reparameterization is a technique to render the sampling operation differentiable. In our approach, we reshape the reparameterization as noise injection operation which involves sampling a latent variable using a deterministic function that incorporates the mean, variance, and an auxiliary variable from a standard distribution. Specifically, we sample two random noise tensors: $\varepsilon_1 \in \mathbb{R}^{B \times C^\mathbf{s}}$ and $\varepsilon_2 \in \mathbb{R}^{B \times C^\mathbf{s}}$ from a normal distribution $N(0, 1)$. We then apply a deterministic function follow the format of reparameterization to inject two random noise tensors respectively into the mean and standard deviation of the downsampled feature representation $\hat{\mathbf{F}}_\mathbf{e}$, this process can be formulated as below:

$$\mu^\mathbf{n} = \mu + \varepsilon_1 * V(\mu), \quad \sigma^\mathbf{n} = \sigma + \varepsilon_2 * V(\sigma) \tag{5}$$

The function $V$ calculates the variance across the batch dimension, and * denotes element-wise multiplication. This reparameterization format allows the network to adaptively learn modifications through its parameters during the noise injection process. Using the altered feature statistics, we restore the normalized feature $\hat{\mathbf{F}}_{\mathbf{z}}$ to obtain the noised feature $\hat{\mathbf{F}}_{\mathbf{n}}$:

$$\hat{\mathbf{F}}_{\mathbf{n}} = \hat{\mathbf{F}}_{\mathbf{z}} * \sigma^{\mathbf{n}} + \mu^{\mathbf{n}} \tag{6}$$

To allocate space for more thorough experiments, we have moved the pseudo-code style pipeline (Algorithm 1), along with the technical explanation and proof part, to Appendix A.3. Please refer to this appendix for comprehensive details. The encoding process in our model integrates varied semantic information from low-level to high-level characteristics during model learning. To enhance performance, noise injection block is applied at each encoding level, for a total of four times. The resultant noised feature $\hat{\mathbf{F}}_{\mathbf{n}}$ possesses significantly altered feature statistics, containing potential domain information from various unseen noise types—crucial for robust model generalization.

## 4 EXPERIMENTS

**Implementation Details.** Our RNINet framework comprises a 4-level encoder-decoder setup. Channel counts across these levels are [64, 128, 256, 512]. In the basic blocks, the convolution layers have a kernel size of 3, while in the noise injection blocks, the downsampling convolution layers feature a kernel size of 2 with a stride of 2. Following the state-of-the-art methodology outlined in MT (Chen et al., 2023), we train our model exclusively with Gaussian noise characterized by a standard deviation of $\sigma = 15$. The testing involves noise types not seen during training. Our training dataset amalgamates DIV2K (Agustsson et al., 2017), Flickr2k (Timofte et al., 2017), BSD400 (Arbeláez et al., 2011), and WED (Ma et al., 2017). All experimental procedures are executed using the PyTorch framework on an RTX 4090 GPU. Training employs the Adam optimizer with $\beta_1 = 0.9$ and $\beta_2 = 0.999$. The learning rate is maintained at $1 \times 10^{-4}$ with a batch size of 8. The training process spans 50,000 iterations, as extending beyond this count might lead to the overfitting of Gaussian noise. For data augmentation, we apply horizontal and vertical flips and extract random $128 \times 128$ patches.

**Testing Noise.** Despite being trained solely on Gaussian noise, the model's generalization capabilities are tested against seven other types of noise: (1) Speckle Noise (2) Salt & Pepper Noise (3) Poisson Noise (4) Image Signal Processing (ISP) Noise (Brooks et al., 2019a). (5) Mixture Noise obtained by mixing the above different types of noise with different levels. The clean images origins from four benchmark datasets: McMaster (Zhang et al., 2011), Kodak24 (Franzen, 1999), CBSD68 (Martin et al., 2001), Urban100 (Huang et al., 2015). We also include two real noise types in our experiments: (6) Monte Carlo Rendering Image Noise and (7) Smartphone Image Noise (Abdelhamed et al., 2018a). Further details on these noise types are provided in the Appendix A.1. For performance evaluation, we employ metrics such as Peak Signal-to-Noise Ratio (PSNR) and Structural Similarity Index Measure (SSIM).

### 4.1 THE GENERALIZATION PERFORMANCE

Our model was trained exclusively on Gaussian noise with $\sigma = 15$, yet it was tested on a range of unencountered non-Gaussian noises to assess its generalization capability. We compared our model with other state-of-the-art models that are not generalizable: SwinIR (Liang et al., 2021), Restormer (Zamir et al., 2022), CODE (Zhao et al., 2023), DRUNet (Zhang et al., 2022), and also with the current state-of-the-art method for generalizable deep image denoising: Masked Training (MT) (Chen et al., 2023), under consistent experimental settings. Additionally, we report on a baseline model where noise injection blocks were substituted with basic blocks. As depicted in Fig. 4 and Tab. 9 in the Appendix, our model consistently outperforms others in terms of generalization performance across various tests. Specifically, in scenarios with speckle noise at $\sigma^2 = 0.02$ and $\sigma^2 = 0.024$, our RNINet registered a PSNR improvement of 1.72 dB and 1.09 dB over MT, and 0.63 dB over the baseline model, respectively. The diminished performance of MT (Chen et al., 2023) can be attributed to its design, which tends to result in over-smoothed images. Although MT's masking strategy yields competitive results in highly noisy conditions (*e.g.*, $\sigma^2 = 0.04$), it falls short in scenarios with moderate noise levels. Besides, our method significant outperforms

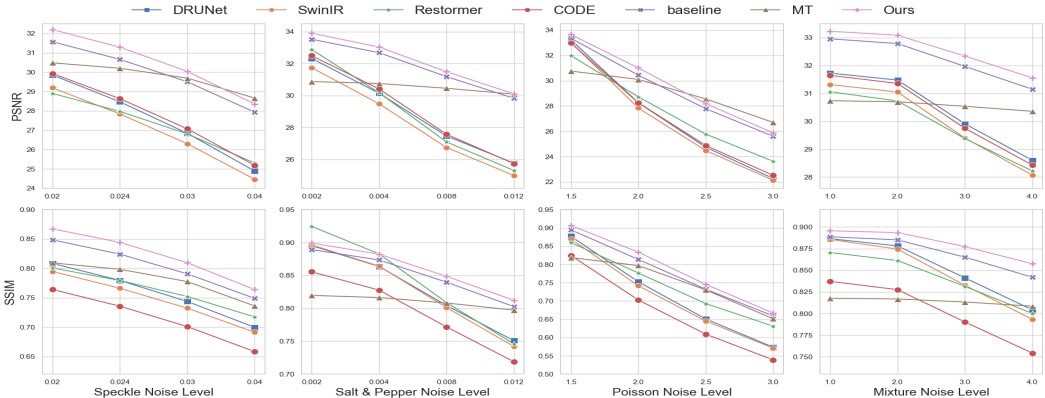

Figure 4: Performance comparisons on four noise types with different levels on the McMaster dataset (Zhang et al., 2011). All models are trained only on Gaussian noise $\sigma = 15$. Our RNINet demonstrates good generalization performance across different noise types. The quantitative results can be found in Tab. 9 in Appendix. For more results on other testsets: Kodak24 (Franzen, 1999), CBSD68 (Martin et al., 2001),Urban100 (Huang et al., 2015), please refer to Appendix A.2.

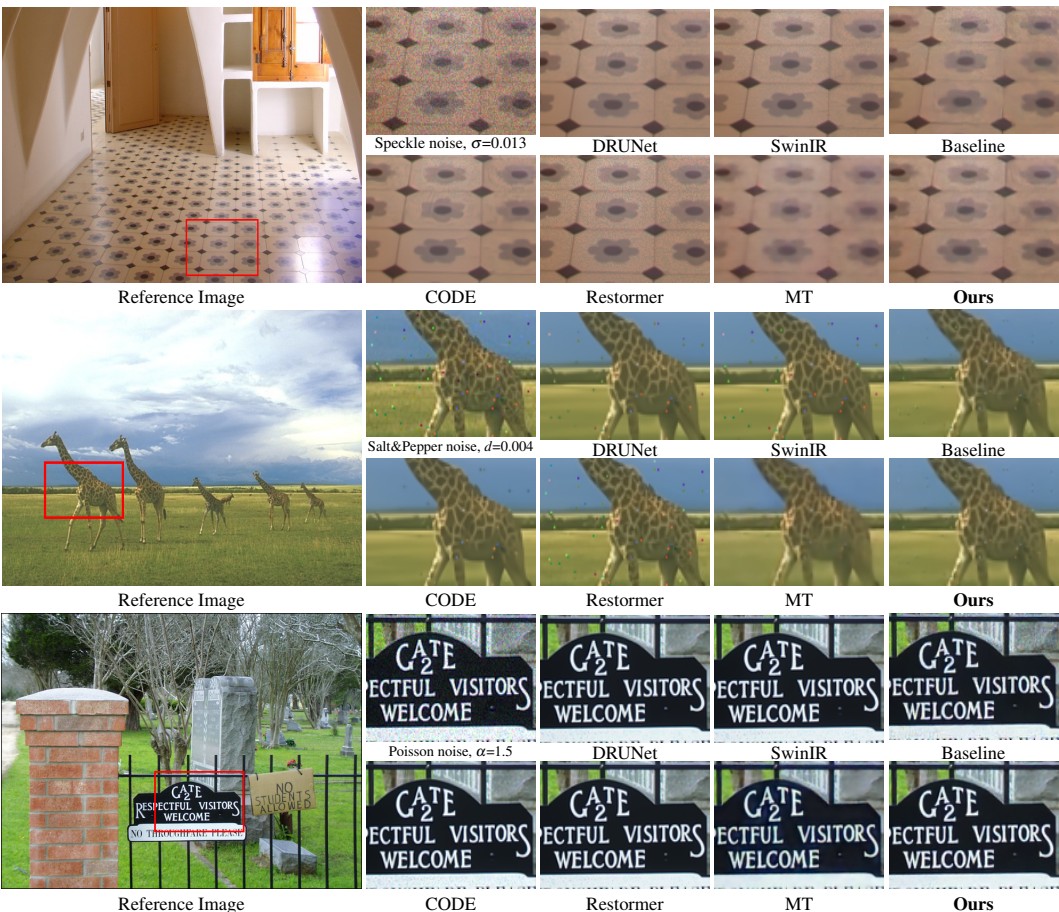

Figure 5: Visual comparisons on out-of-distribution noises. Our RNINet is trained only on Gaussian noise but can generalize well to other unseen noises. Compared with MT (Chen et al., 2023) which generates over-smoothed content, our model can preserve more details, therefore leading to higher PSNR and SSIM in testsets.

MT in handling mixture noises, which are more complicated and representative in real application environments. Visual comparisons in Fig. 5 further demonstrate that our model yields comparable denoising outcomes compared to both non-generalizable models and the generalizable model.

## 4.2 EVALUATION ON MONTE CARLO RENDERING NOISE REMOVAL

We extend our evaluation to include the removal of noise from Monte Carlo rendering, consistent with MT (Chen et al., 2023). Monte Carlo denoising is a critical component of the rendering process, especially given the prevalent use of Monte Carlo rendering algorithms in the industry (Burley et al., 2018; Christensen et al., 2018; Kulla et al., 2018). We utilize the test dataset proposed by (Firmino et al., 2022) and convert the raw dataset to the sRGB color space for our Monte Carlo rendered image denoising experiments. The test images were rendered at varying levels of samples per pixel (spp): 256 spp, 128 spp, and 64 spp. Notably, the lower the spp, the higher the noise intensity in the images. Tab. 1 and Fig. 6 display the denoising results. Our method demonstrates superior performance across all settings (256 spp, 128 spp, and 64 spp) and produces visually more appealing images with reduced noise and enhanced details.

| Method | 256 Samples Per Pixel | | 128 Samples Per Pixel | | 64 Samples Per Pixel | |
|---|---|---|---|---|---|---|
| | PSNR↑ | SSIM↑ | PSNR↑ | SSIM↑ | PSNR↑ | SSIM↑ |
| DRUNet (Zhang et al., 2022) | 33.12 | 0.8656 | 29.78 | 0.7882 | 26.70 | 0.7068 |
| SwinIR (Liang et al., 2021) | 33.09 | 0.8566 | 29.70 | 0.7767 | 26.54 | 0.6948 |
| Restormer (Zamir et al., 2022) | 28.48 | 0.7342 | 27.08 | 0.6578 | 25.76 | 0.6620 |
| CODE (Zhao et al., 2023) | 31.98 | 0.7815 | 29.14 | 0.6995 | 26.28 | 0.6138 |
| Baseline | 32.69 | 0.8630 | 29.74 | 0.7940 | 27.12 | 0.7134 |
| MT (Chen et al., 2023) | 30.32 | 0.7769 | 28.87 | 0.7219 | 26.78 | 0.6349 |
| **Ours** | **33.36** | **0.8760** | **30.34** | **0.8056** | **27.38** | **0.7244** |

Table 1: Quantitative comparisons on Monte Carlo rendering image denoising.

## 4.3 EVALUATION ON ISP NOISE REMOVAL

Aligned with the current state-of-the-art, MT (Chen et al., 2023), we have conducted evaluations of our model on ISP noise removal to assess its practical significance. Utilizing the systematic approach proposed by Brooks et al. (Brooks et al., 2019a) for generating realistic raw data with ISP noise, we applied the default parameter settings from their research to synthesize ISP noise using the McMaster dataset (Zhang et al., 2011) for our experiments. The comparative results, presented in Tab. 2, demonstrate that our method not only outperforms other models but also surpasses the MT (Chen et al., 2023) benchmark by 1 dB in PSNR, despite a minor reduction in SSIM. The visualizations shown in Fig. 8 in Appendix further highlight our model's capability to preserve image detail while effectively mitigating ISP noise, confirming its robustness in this domain.

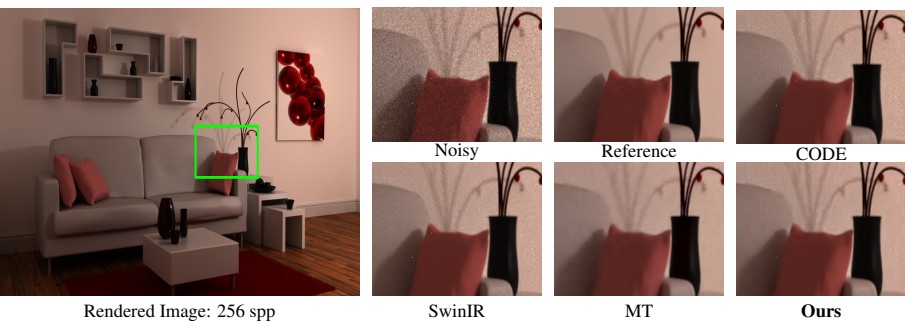

Figure 6: Visual comparisons on Monte Carlo rendering noise removal.

## 4.4 EVALUATION ON SMARTPHONE IMAGE NOISE REMOVAL

In addition to Monte Carlo rendering noise, in our study, we address another real world noise type:

Smartphone Image Noise. The Smartphone Image Denoising Dataset (SIDD) (Abdelhamed et al., 2018a) includes images from 10 different scenes captured under various lighting conditions using five representative cameras. Testing the generalization performance of models on the SIDD validation dataset is particularly challenging due to the complexity of the noise and the requirement that models must not have prior information about the noise distributions in SIDD for a fair generalization test. As shown in Tab. 3, our method achieves significant improvements in PSNR (+0.24 dB than MT (Chen et al., 2023)) over competing methods even under these stringent conditions. The visual results, presented in Fig. 9 in Appendix, demonstrate that while the MT method tends to oversmooth content, resulting in unclear edges, neither CODE nor SwinIR effectively reduce the appearance of black cyan noise spots, but our model can preserve image detail while effectively reducing noise. It is crucial to note that we had no access to the images from the training or testing portions of the SIDD prior to testing, meaning we lacked prior information about the image content and noise distribution within the dataset. Although some supervised methods (Chen et al., 2022a; Yue et al., 2020; Jang et al., 2024) have demonstrated superior results on SIDD, their models can not generalize as effectively to other noise types as our RNINet does due to that they are still overfitting the noise in SIDD dataset.

| Method | Synthetic ISP Noise | |
| --- | --- | --- |
| | PSNR↑ | SSIM↑ |
| DRUNet (Zhang et al., 2022) | 31.01 | 0.8033 |
| SwinIR (Liang et al., 2021) | 31.09 | 0.7968 |
| Restormer (Zamir et al., 2022) | 26.02 | 0.6762 |
| CODE (Zhao et al., 2023) | 30.59 | 0.7908 |
| Baseline | 30.99 | 0.8129 |
| MT (Chen et al., 2023) | 30.15 | 0.8232 |
| **Ours** | **31.15** | **0.8195** |

Table 2: Quantitative comparisons on synthetic ISP noise removal.

| Method | Smartphone Image Noise | |
| --- | --- | --- |
| | PSNR↑ | SSIM↑ |
| DRUNet (Zhang et al., 2022) | 28.09 | 0.5726 |
| SwinIR (Liang et al., 2021) | 27.62 | 0.5604 |
| Restormer (Zamir et al., 2022) | 22.54 | 0.3700 |
| CODE (Zhao et al., 2023) | 26.81 | 0.5186 |
| Baseline | 28.64 | 0.5953 |
| MT (Chen et al., 2023) | 28.66 | 0.6044 |
| **Ours** | **28.90** | **0.6041** |

Table 3: Quantitative comparisons on smartphone image noise removal.

## 4.5 IN DISTRIBUTION PERFORMANCE COMPARISON

Our method has shown promising results on unseen noise types, highlighting its excellent generalization capabilities, it is equally important to assess performance on in-distribution noise. Therefore, we conducted a comparative analysis with the state-of-the-art generalizable method MT (Chen et al., 2023), which is also trained on single Gaussian noise with $\sigma = 15$. Following the Gaussian denoise benchmark criteria outlined in (Liang et al., 2021; Zamir et al., 2022; Zhao et al., 2023), we tested both MT and our RNINet on four testsets: McMaster (Zhang et al., 2011), Kodak24 (Franzen, 1999), CBSD68 (Martin et al., 2001), Urban100 (Huang et al., 2015). As illustrated in Tab. 4, our method significantly outperforms MT in both PSNR and SSIM, with improvements in PSNR ranging from a minimum of 2.46 dB to a maximum of 3.46 dB. MT's underperformance in in-distribution conditions can be attributed to its masking operation, which tends to overly smooth image content to achieve better generalization. In contrast, our method not only delivers good generalization performance but also excels in in-distribution denoising scenarios, further demonstrating the robustness of our noise removal approach.

## 4.6 MACS, GPU MEMORY USAGE AND RUNTIME

Resource efficiency is crucial when deploying generalizable denoising models in real-world applications. Therefore, we report on MACs, GPU memory cost, and runtime comparisons between MT (Chen et al., 2023) and our RNINet. As indicated in Tab. 5, RNINet demonstrates significantly lower MACs, reduced GPU memory usage, and faster runtime, attributable to our model's simplified architectural design. Unlike MT, which employs a complex masked swinir structure leading to higher resource consumption and longer runtimes, RNINet offers superior resource cost efficiency alongside enhanced general denoising performance. This makes RNINet particularly suitable for applications requiring efficient operation without compromising on denoising quality.

| | MT | | Ours | |
|---|---|---|---|---|
| Dataset | PSNR↑ | SSIM↑ | PSNR↑ | SSIM↑ |
| McMaster | 30.85 | 0.8325 | **34.05**$_{\uparrow 3.20}$ | **0.9122**$_{\uparrow 0.0797}$ |
| Kodak24 | 31.65 | 0.8836 | **34.11**$_{\uparrow 2.46}$ | **0.9128**$_{\uparrow 0.0292}$ |
| CBSD68 | 31.00 | 0.8880 | **33.50**$_{\uparrow 2.50}$ | **0.9217**$_{\uparrow 0.0337}$ |
| Urban100 | 29.51 | 0.8992 | **32.97**$_{\uparrow 3.46}$ | **0.9310**$_{\uparrow 0.0318}$ |

Table 4: Quantitative comparisons on in-distribution denoising performance. Our method significantly outperforms MT (Chen et al., 2023) in both PSNR and SSIM across four benchmark datasets. The ↑ indicates the absolute improvement over MT, highlighting the robust in-distribution denoising capability of our approach.

| Metrics | MT | **Ours** |
|---|---|---|
| MACs | 51.6G | **44.7G**$_{\downarrow 0.87\times}$ |
| GPU Memory Usage | 0.90G | **0.62G**$_{\downarrow 0.69\times}$ |
| Runtime | 1.04s | **0.10s**$_{\downarrow 0.10\times}$ |

Table 5: Quantitative comparisons on MACs, GPU memory usage and runtime. MACs and GPU memory usage were tested on images of size $256 \times 256$. Runtime was tested by calculating the average inference time per image in McMaster testset. The ↓ denotes the relative rate compared with MT, highlighting the efficiency of our method.

### 4.7 GENERALIZATION ANALYSIS

Following (Kong et al., 2022; Wang et al., 2024), we utilize the Deep Degradation Representation (DDR) introduced by (Liu et al., 2021) and visualize it in Fig. 7. DDR enables us to probe the network's generalization capabilities by analyzing model behavior across different degradations. In our cases, each point corresponds to an input image, with different colors denoting various noise types. For instance, in the cases of SwinIR and Restormer, it is noticeable that images with identical noise types tend to cluster together, suggesting that these models may encode noise-specific information, potentially leading to an overfitting issue. Nevertheless, the points in MT and our method are clustered based on their content rather than noise type. (Liu et al., 2021) also employs the Calinski-Harabaz Index (CHI) score for a more quantitative analysis of clustering efficiency, where a lower CHI score indicates better cluster separation and, consequently, superior generalization ability. Notably, both MT and our method achieve very low CHI scores; however, our method's CHI score is still significantly lower than that of MT, which underscores our model's enhanced generalization performance.

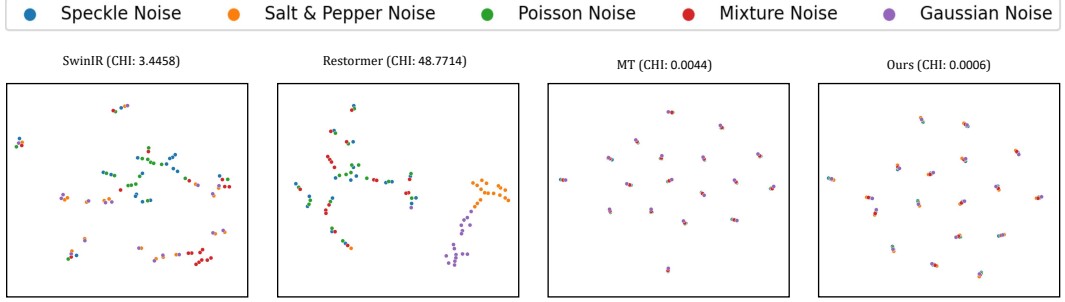

Figure 7: Visualization of the DDR clusters for generalization analysis. A lower CHI score demonstrates better cluster separation, which in turn suggests superior generalization capability.

## 5 CONCLUSION

In this work, we introduce RNINet, a novel architecture designed for generalizable deep image denoising. RNINet incorporates a streamlined encoder-decoder framework with noise injection blocks, addressing the prevalent issue of over-smoothing observed in the existing MT method, while also enhancing both efficiency and overall performance. Our approach leverages the insight that feature statistics, such as the mean and variance of a denoising model, shift significantly in response to different noise types used during training. To exploit this, we introduce a noise injection block within our framework that injects random noise into the feature statistics, significantly improving generalization across unseen noise types. Extensive experimental results demonstrate that RNINet not only surpasses the state-of-the-art MT method in terms of denoising effectiveness and computational efficiency but also achieves inference speeds up to ten times faster. This study not only advances the field of generalizable deep image denoising but also paves the way for future research into broader applications of robust noise-handling models.

ACKNOWLEDGMENTS

This research was supported in part by JSPS KAKENHI Grant Numbers 24KK0209, 24K22318, 22H00529, JST-Mirai Program JPMJMI23G1 and JST SPRING Grant Number JPMJSP2108.

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

# A APPENDIX

## A.1 DETAILS OF TEST NOISE

In this section, we provide details on the test noises used in our evaluations:

**Speckle Noise**: Speckle noise is a granular noise that inherently degrades the quality of images produced by coherent imaging systems such as laser, synthetic aperture radar, and ultrasound. It arises from the random interference of coherent waves scattered by the surface roughness of the target. Following the method outlined in MT (Chen et al., 2023), we use the *imnoise* function in MATLAB to generate speckle noise. The noise level is controlled by the parameter $\sigma^2$.

**Salt & Pepper Noise**: Salt-and-pepper noise is a type of impulse noise characterized by randomly occurring white and black pixels in an image, typically caused by errors in data transmission or malfunctioning camera sensors. As described in MT (Chen et al., 2023), we use the *imnoise* function in MATLAB to generate salt & pepper noise, with the parameter $d$ controlling the noise level.

**Poisson Noise**: Also known as shot noise, Poisson noise occurs in systems where the signal consists of statistically independent discrete events. This type of noise follows a Poisson distribution and is commonly observed in photon counting processes in optical systems. Following MT (Chen et al., 2023), we amplified the noise using different scaling factors $\alpha$ according to the equation $J = I + n \cdot \alpha$, where $n$ represents the generated Poisson noise, and $\alpha$ is the scaling factor.

**Image Signal Processing (ISP) Noise**: ISP noise encompasses various types of noise introduced during the process of capturing and converting signals from an image sensor into digital images. This includes noise from sensor readout, fixed pattern noise, dark current noise, and other electronic or thermal interferences. We use the code provided by Brooks et al. (Brooks et al., 2019a) and apply the default settings to convert RGB images to raw format, add ISP noise, and then convert them back to RGB format.

**Monte Carlo Rendered Image Noise**: Monte Carlo rendered image noise arises in images generated by Monte Carlo rendering techniques, due to the stochastic nature of sampling. Insufficient samples can lead to visible graininess or speckles. We utilize the test dataset proposed by Firmino et al. (Firmino et al., 2022) and convert the raw dataset to the sRGB color space using the code from another study (Han et al., 2023). Unlike MT (Chen et al., 2023), we did not use the tonemapping function when converting raw images to RGB images due to the unavailability of configuration parameters and codes for tonemapping in their work.

**Smartphone Image Noise**: We utilized the Smartphone Image Denoising Dataset (SIDD) (Abdelhamed et al., 2018a), which comprises images from 10 different scenes captured under various lighting conditions using five representative cameras, to evaluate the generalization capabilities of our RNINet and other methods. We processed the SIDD evaluation data using data preparation codes from NAFNet (Chen et al., 2022a) and directly tested several benchmark methods trained exclusively on Gaussian noise with $\sigma = 15$. It is important to note that we did not have access to any images from the training and testing datasets before testing, which means we had no prior information about the image content and noise distribution within that dataset. While some supervised methods have obtained comparable results on the SIDD evaluation, they can not generalize as well to other noise types as our RNINet does.

**Mixture Noise**: We simulate real-world noise scenarios by mixing different types of noise with varying intensities, resulting in four levels of mixture noise. The order of noise addition is speckle noise (variance $\sigma_{s1}^2$), salt & pepper noise (density $d$), and Poisson noise (scale $\alpha$). Unlike MT (Chen et al., 2023), we did not add Gaussian noise, as it is a in-distribution noise that reduces denoising difficulty. The four levels are as follows:

- Level 1: $\sigma_{s1}^2 = 0.003$, $d = 0.002$, $\alpha = 1$

- Level 2: $\sigma_{s1}^2 = 0.004$, $d = 0.002$, $\alpha = 1$

- Level 3: $\sigma_{s1}^2 = 0.006$, $d = 0.003$, $\alpha = 1$

- Level 4: $\sigma_{s1}^2 = 0.008$, $d = 0.004$, $\alpha = 1$

By providing these detailed descriptions and parameters, we aim to ensure the reproducibility and clarity of our noise generation processes.

## A.2 ADDITIONAL RESULTS AND ABLATION EXPERIMENTS

**Additional Ablation Experiments.** As detailed in Section. 4 and illustrated in our experiment tables, the most important ablation study reported involves a baseline model where noise injection blocks were replaced with basic blocks. Regarding more ablation studies, we adjusted the intensity of random noise injected into feature statistics by applying a scaling factor. The results are presented in Tab. 6. We found that the original scale (no scaling) delivers the best performance compared to other methods. We believe this superior performance is due to the scaling operation disrupting the standard values in the tensors sampled from a Gaussian distribution, thereby degrading the overall performance.

| Speckle Noise Scale | $\sigma^2 = 0.02$ PSNR↑/SSIM↑ | $\sigma^2 = 0.024$ PSNR↑/SSIM↑ | $\sigma^2 = 0.03$ PSNR↑/SSIM↑ | $\sigma^2 = 0.04$ PSNR↑/SSIM↑ |
|---|---|---|---|---|
| 1.2 | 31.06/0.8310 | 29.96/0.8032 | 28.62/0.7685 | 26.99/0.7250 |
| 1.1 | 31.20/0.8411 | 30.20/0.8136 | 28.87/0.7784 | 27.18/0.7342 |
| 0.9 | 31.30/0.8335 | 30.25/0.8076 | 28.88/0.7738 | 27.15/0.7295 |
| 0.8 | 31.44/0.8469 | 30.24/0.8189 | 28.84/0.7829 | 27.02/0.7369 |
| **1.0 (Original)** | **32.20/0.8674** | **31.29/0.8443** | **30.04/0.8097** | **28.35/0.7639** |

Table 6: We adjusted the intensity of random noise injected into feature statistics by applying a scaling factor. The experiments demonstrate no scaling delivers the best performance compared to other methods. We believe this superior performance is due to the scaling operation disrupting the standard values in the tensors sampled from a Gaussian distribution, thereby degrading the overall performance.

**Additional Comparison with Zero-Shot Real-World Denoisers.** LAN (Kim et al., 2024) is a recent approach designed to bridge the noise distribution gap across various real-world denoising datasets, thereby enhancing generalization capabilities. To further validate the effectiveness of our method under zero-shot real-world denoising conditions, we included comparisons with two zero-shot denoisers from LAN (Kim et al., 2024), alongside the previous state-of-the-art (SOTA) method, MT. Following the experimental setup in LAN, we processed the PolyU (Xu et al., 2018) and Nam (Nam et al., 2016) testsets. The results are presented in Tab. 7. We believe these findings provide additional evidence of our method's strong generalization capability in addressing zero-shot real-world denoising conditions.

| Dataset | PolyU (Xu et al., 2018) PSNR↑ | SSIM↑ | Nam (CC) (Nam et al., 2016) PSNR↑ | SSIM↑ |
|---|---|---|---|---|
| ZS-Denoiser (ZS-N2N) (Kim et al., 2024) | 31.47 | 0.8750 | 34.47 | 0.9020 |
| ZS-Denoiser (Nbr2Nbr) (Kim et al., 2024) | 32.93 | 0.9110 | 35.39 | 0.9230 |
| MT (Chen et al., 2023) | 33.96 | 0.9260 | 33.20 | 0.9209 |
| **RNINet (Ours)** | **37.59** | **0.9551** | **37.15** | **0.9525** |

Table 7: Additional results on PolyU (Xu et al., 2018) and Nam (Nam et al., 2016) datasets, compared with MT and two zero-shot denoisers introduced in LAN (Kim et al., 2024).

**Additional Results on Four Benchmark Datasets.** In this section, we first present quantitative results on four benchmark datasets: McMaster (Zhang et al., 2011), CBSD68 (Martin et al., 2001), Kodak24 (Franzen, 1999), and Urban100 (Huang et al., 2015). As shown in Tab. 9, Tab. 10, Tab. 11, and Tab. 12, our RNINet outperforms other state-of-the-art methods in most cases. Specifically, RNINet achieves significantly better performance than other methods for mixture noises in all settings, which are more complicated and representative in real application environments. Besides these significant improvements, our RNINet also demonstrates computational efficiency, with lower MACs and GPU memory usage, RNINet achieves up to 10 times faster inference speeds compared to

MT (Chen et al., 2023) as indicated in Tab. 5, making it particularly suitable for applications requiring efficient operation without compromising on denoising quality.

## A.3 THEORETICAL EXPLANATION AND PROOF

In this section, we provide a theoretical explanation and proof of our random noise injection strategy, demonstrating why it can improve generalization capability on unseen noise types. This process consists of two steps. In the forward step, we prove that features generated from images with different noise distributions can form a normal distribution. In the backward step, we explain how altered features noised by random tensors sampled from a normal distribution can be mapped through a nonlinear neural network to match features from images with unseen noise types.

---

**Algorithm 1** Pipeline and Pseudo-code

---

**Input:** Normal feature $\hat{\mathbf{F}}_{\mathbf{e}}$ from basic block encoder
**Output:** Noised feature $\hat{\mathbf{F}}_{\mathbf{n}}$ through noise injection
**Define:** $\hat{\mathbf{F}}_{\mathbf{e}} \in \mathbb{R}^{B \times H^s \times W^s \times C^s}$, $\mu$ represents the mean and $\sigma$ represents the standard deviation.
$V$ calculates the variance across the batch dimension, * denotes element-wise multiplication.

1: **Prepare**
2:     Calculate the channel-wise mean $\mu \in \mathbb{R}^{B \times C^s}$ by Equation 2.
3:     Calculate the channel-wise standard deviation $\sigma \in \mathbb{R}^{B \times C^s}$ by Equation 3.
4: **Then**
5:     # Obtain normalized feature
6:     $\hat{\mathbf{F}}_{\mathbf{z}} = (\hat{\mathbf{F}}_{\mathbf{e}} - \mu)/\sigma$
7:     # Sample tensors from normal distribution
8:     $\varepsilon_1 \sim N(0,1), \quad \varepsilon_2 \sim N(0,1)$
9:     # Inject noise with deterministic function
10:     Noised $\mu \rightarrow \mu^{\mathbf{n}} = \mu + \varepsilon_1 * V(\mu)$
11:     Noised $\sigma \rightarrow \sigma^{\mathbf{n}} = \sigma + \varepsilon_2 * V(\sigma)$
12:     # Alter the normalized feature
13:     $\hat{\mathbf{F}}_{\mathbf{n}} = \hat{\mathbf{F}}_{\mathbf{z}} * \sigma^{\mathbf{n}} + \mu^{\mathbf{n}}$
14: **Return** Noised feature $\hat{\mathbf{F}}_{\mathbf{n}}$

---

**Forward:** Suppose the neural network $N$ is a complex nonlinear function, with $f_i = N(x_i)$ representing the feature map of an intermediate layer after input $x_i$ passes through the neural network. Assume that for different inputs $x_i$, their noise distributions follow different distributions such as speckle noise distribution, salt & pepper noise distribution, poisson noise distribution, etc.

According to the Lindeberg-Feller Central Limit Theorem, even if these $x_i$ have different noise distributions, the sum or average of $f_i$ will converge to a normal distribution if certain requirements are met. Due to that $x_i$ are independent but differently distributed, therefore $f_i$ are also independent but differently distributed, a large number of $f_i$ can form a normal distribution. The examination of the four conditions for the Lindeberg-Feller Central Limit Theorem is as follows:

1. **Independence**: ✓, $f_i$ is independent, with each $f_i$ generated from $x_i$ following a different noise distribution.

2. **Existence of Expectation and Variance**: ✓, the expectation $\mu_i$ and variance $\sigma_i^2$ of $f_i$ exist, as we have calculated the feature statistics before.

3. **Normalization Condition**: ✓, $\sum_{i=1}^{n} \sigma_i^2 \rightarrow \infty$, the number $n$ of training sample is very large.

4. **Lindeberg Condition**: ✓, for any $\varepsilon > 0$,

$$\frac{1}{\sum_{i=1}^{n} \sigma_i^2} \sum_{i=1}^{n} E\left[(f_i - \mu_i)^2 \cdot I\left(|f_i - \mu_i| \geq \varepsilon \sqrt{\sum_{i=1}^{n} \sigma_i^2}\right)\right] \rightarrow 0$$

**Backward:** During model training, we only accept input images $x_i^{\alpha}$ with Gaussian noise at a level of 15, with the feature map represented as $f_i^{\alpha}$. As we have proven above, a large number of $f_i$ from different noise distribution can form a normal distribution. Thus, in return, we sample random noise tensors $\varepsilon_1$ and $\varepsilon_2$ from a normal distribution $N(0, 1)$, and inject them into the feature statistics ($\mu_i^{\alpha}$ and $\sigma_i^{\alpha}$) of feature map $f_i^{\alpha}$.

1. Through a proper nonlinear mapping function, we can map the noised $f_i^\alpha$ by $\varepsilon_1$ and $\varepsilon_2$ to match the unseen $f_i$, for improved generalization capability.

2. We use reparameterization to make this noise injection differentiable, which enables the neural network to automatically learn suitable nonlinear mapping functions for $f_i^\alpha \to f_i$ .

## A.4 Primary Exploration Using Multiple Noise Types for Training

The original setup for this task follows the previous SOTA method MT (Chen et al., 2023), which involves training solely on Gaussian noise, with the aim of generalizing to other unseen noise types. This setup simplifies data construction and focuses primarily on developing a robust model architecture. However, we recognize that the community may be interested in results obtained when the model is trained not only on Gaussian noise but also on other noise types (e.g., Poisson noise).

To address this, we recently conducted preliminary exploratory experiments. Specifically, we trained our RNINet model on Gaussian noise initially, followed by training on Poisson noise. This version is referred to as RNINet-g2p. The total number of iterations for RNINet-g2p was kept identical to that used for training RNINet exclusively on Gaussian noise (referred to as RNINet-original). While we observed minor improvements in a few cases, the overall results showed a significant performance drop, as presented in Tab. 8.

In conclusion, we believe that training solely on Gaussian noise yields overall satisfactory results. However, there remains considerable potential for further exploration into the optimal combination of noise types during training to achieve improved performance. This is an intriguing and open research question that we are actively considering. We are optimistic that future studies will investigate this phenomenon more thoroughly, both from theoretical and experimental perspectives. We hope our findings can provide valuable insights and serve as a foundation for future research in this direction.

| Speckle Noise | $\sigma^2 = 0.02$ | | $\sigma^2 = 0.024$ | | $\sigma^2 = 0.03$ | | $\sigma^2 = 0.04$ | |
| --- | --- | --- | --- | --- | --- | --- | --- | --- |
| Method | PSNR↑ | SSIM↑ | PSNR↑ | SSIM↑ | PSNR↑ | SSIM↑ | PSNR↑ | SSIM↑ |
| RNINet-g2p | 31.42 | 0.8425 | 30.39 | 0.8153 | 29.09 | 0.7801 | 27.48 | 0.7341 |
| RNINet-original | 32.20 | 0.8674 | 31.29 | 0.8443 | 30.04 | 0.8097 | 28.35 | 0.7639 |
| Salt & Pepper | $d = 0.002$ | | $d = 0.004$ | | $d = 0.008$ | | $d = 0.012$ | |
| Method | PSNR↑ | SSIM↑ | PSNR↑ | SSIM↑ | PSNR↑ | SSIM↑ | PSNR↑ | SSIM↑ |
| RNINet-g2p | 34.54 | 0.9290 | 32.93 | 0.9042 | 30.65 | 0.8554 | 29.00 | 0.8075 |
| RNINet-original | 33.91 | 0.8992 | 33.04 | 0.8824 | 31.50 | 0.8486 | 30.11 | 0.8123 |
| Poisson Noise | $\alpha = 1.5$ | | $\alpha = 2$ | | $\alpha = 2.5$ | | $\alpha = 3$ | |
| Method | PSNR↑ | SSIM↑ | PSNR↑ | SSIM↑ | PSNR↑ | SSIM↑ | PSNR↑ | SSIM↑ |
| RNINet-g2p | 32.72 | 0.8747 | 29.36 | 0.7658 | 26.80 | 0.6730 | 24.81 | 0.5970 |
| RNINet-original | 33.65 | 0.9067 | 31.01 | 0.8337 | 28.19 | 0.7457 | 25.87 | 0.6673 |
| Mixture Noise | Level 1 | | Level 2 | | Level 3 | | Level 4 | |
| Method | PSNR↑ | SSIM↑ | PSNR↑ | SSIM↑ | PSNR↑ | SSIM↑ | PSNR↑ | SSIM↑ |
| RNINet-g2p | 32.79 | 0.8960 | 32.51 | 0.8880 | 31.24 | 0.8551 | 30.22 | 0.8227 |
| RNINet-original | 33.23 | 0.8956 | 33.09 | 0.8935 | 32.34 | 0.8774 | 31.56 | 0.8575 |
| | Synthetic ISP Noise | | Monte Carlo Rendering Noise | | | | | |
| Method | PSNR↑ | SSIM↑ | PSNR↑ (256 pp) | SSIM↑ (256 pp) | PSNR↑ (128 pp) | SSIM↑ (128 pp) | PSNR↑ (64 pp) | SSIM↑ (64 pp) |
| RNINet-g2p | 30.88 | 0.7963 | 31.74 | 0.8240 | 29.18 | 0.7477 | 26.47 | 0.6668 |
| RNINet-original | 31.15 | 0.8195 | 33.36 | 0.8760 | 30.34 | 0.8056 | 27.38 | 0.7244 |

Table 8: Additional comparisons are provided by training our RNINet model on Gaussian noise first, followed by training on Poisson noise. This approach is referred to as RNINet-g2p. The total number of iterations for RNINet-g2p was kept identical to the iterations used for training RNINet exclusively on Gaussian noise (referred to as RNINet-original).

## A.5 Visualizations

Here, we present additional visualizations referenced in the main paper. For detailed explanations and analysis, please refer to the corresponding sections in the main text.

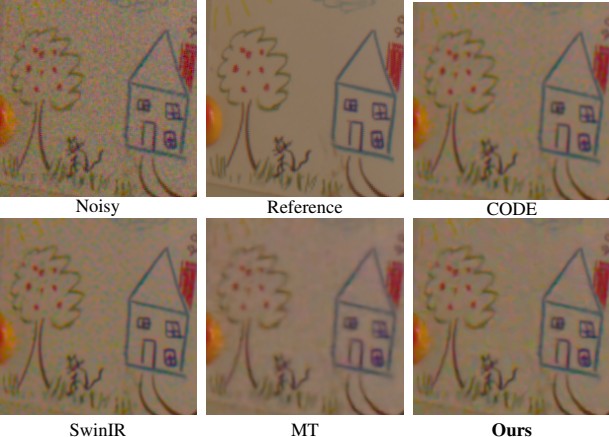

Figure 8: Visual comparisons on synthetic ISP noise removal. Our method can remove noise and preserve more details, therefore leading to higher PSNR and SSIM in testsets.

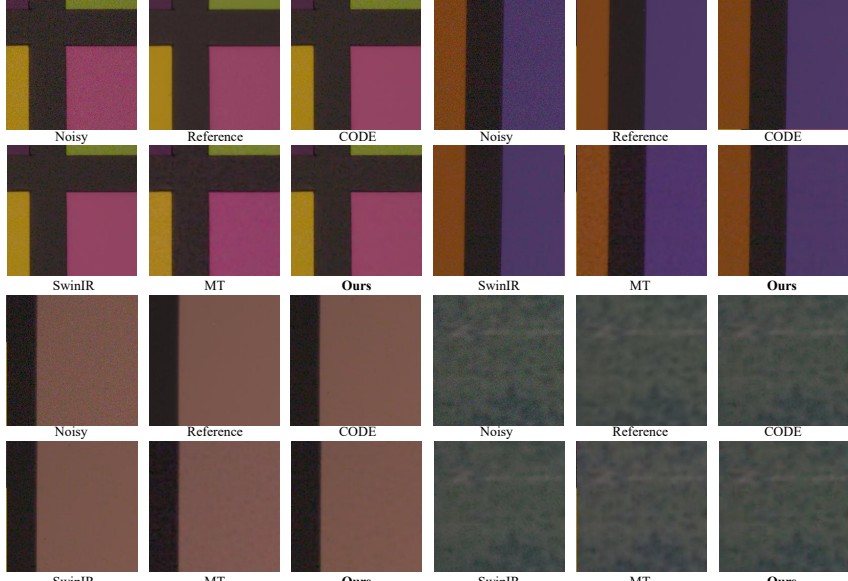

Figure 9: Visual comparisons on smartphone image noise removal. Our method can remove noise and preserve more details, therefore leading to higher PSNR and SSIM in testsets.

## A.6 Limitation and Future Direction

While our method has achieved promising results, enhancing both efficiency and overall performance, there are still areas that warrant further exploration. Compared to MT (Chen et al., 2023), our approach generally performs better in most scenarios with 10 times faster inference speeds, particularly in handling mixture noise which is more complex and representative in real cases. However, MT (Chen et al., 2023) exhibits competitive performance under super high-intensity noise conditions due to its stronger smoothing capabilities. Although such high levels of noise (somewhat rare and peculiar) are unlikely to be encountered in real-world conditions, where noise is typically not as artificially extreme as in MT's setting (*i.e.* poisson noise multiplied by the scale factor 2/2.5/3), it is still valuable to investigate how our method can further adapt to super-high noise intensities. Specifically, improving our method's ability to provide stronger smoothing under extremely high noise conditions without excessively smoothing images in moderate noise scenarios remains an important area for future research.

| Speckle Noise | $\sigma^2 = 0.02$ | | $\sigma^2 = 0.024$ | | $\sigma^2 = 0.03$ | | $\sigma^2 = 0.04$ | |
|---|---|---|---|---|---|---|---|---|
| Method | PSNR↑ | SSIM↑ | PSNR↑ | SSIM↑ | PSNR↑ | SSIM↑ | PSNR↑ | SSIM↑ |
| DRUNet | 29.85 | 0.8085 | 28.47 | 0.7795 | 26.84 | 0.7437 | 24.89 | 0.6996 |
| SwinIR | 29.19 | 0.7948 | 27.84 | 0.7662 | 26.30 | 0.7323 | 24.46 | 0.6912 |
| Restormer | 28.90 | 0.8008 | 27.97 | 0.7794 | 26.81 | 0.7522 | 25.30 | 0.7174 |
| CODE | 29.93 | 0.7641 | 28.63 | 0.7357 | 27.07 | 0.7009 | 25.18 | 0.6583 |
| baseline | 31.57 | 0.8486 | 30.66 | 0.8243 | 29.50 | 0.7907 | 27.93 | 0.7489 |
| MT | 30.48 | 0.8094 | 30.20 | 0.7985 | 29.69 | 0.7776 | 28.66 | 0.7360 |
| **Ours** | **32.20** | **0.8674** | **31.29** | **0.8443** | **30.04** | **0.8097** | **28.35** | **0.7639** |
| Salt & Pepper | $d = 0.002$ | | $d = 0.004$ | | $d = 0.008$ | | $d = 0.012$ | |
| Method | PSNR↑ | SSIM↑ | PSNR↑ | SSIM↑ | PSNR↑ | SSIM↑ | PSNR↑ | SSIM↑ |
| DRUNet | 32.33 | 0.8953 | 30.15 | 0.8637 | 27.47 | 0.8044 | 25.77 | 0.7508 |
| SwinIR | 31.75 | 0.8964 | 29.49 | 0.8642 | 26.75 | 0.8012 | 25.00 | 0.7420 |
| Restormer | 32.89 | 0.9246 | 30.18 | 0.8828 | 27.10 | 0.8088 | 25.31 | 0.7459 |
| CODE | 32.50 | 0.8555 | 30.42 | 0.8277 | 27.58 | 0.7718 | 25.74 | 0.7188 |
| baseline | 33.53 | 0.8893 | 32.69 | 0.8735 | 31.19 | 0.8398 | 29.83 | 0.8026 |
| MT | 30.88 | 0.8197 | 30.76 | 0.8164 | 30.46 | 0.8081 | 30.09 | 0.7973 |
| **Ours** | **33.91** | **0.8992** | **33.04** | **0.8824** | **31.50** | **0.8486** | **30.11** | **0.8123** |
| Poisson Noise | $\alpha = 1.5$ | | $\alpha = 2$ | | $\alpha = 2.5$ | | $\alpha = 3$ | |
| Method | PSNR↑ | SSIM↑ | PSNR↑ | SSIM↑ | PSNR↑ | SSIM↑ | PSNR↑ | SSIM↑ |
| DRUNet | 33.24 | 0.8773 | 28.26 | 0.7527 | 24.74 | 0.6510 | 22.27 | 0.5735 |
| SwinIR | 33.04 | 0.8697 | 27.84 | 0.7418 | 24.48 | 0.6453 | 22.13 | 0.5712 |
| Restormer | 31.99 | 0.8601 | 28.73 | 0.7768 | 25.78 | 0.693 | 23.63 | 0.6312 |
| CODE | 32.98 | 0.8247 | 28.26 | 0.7029 | 24.88 | 0.6094 | 22.52 | 0.5389 |
| baseline | 33.36 | 0.8949 | 30.44 | 0.8139 | 27.79 | 0.7324 | 25.60 | 0.6593 |
| MT | 30.76 | 0.8187 | 30.10 | 0.7972 | 28.56 | 0.7306 | 26.70 | 0.6514 |
| **Ours** | **33.65** | **0.9067** | **31.01** | **0.8337** | **28.19** | **0.7457** | **25.87** | **0.6673** |
| Mixture Noise | Level 1 | | Level 2 | | Level 3 | | Level 4 | |
| Method | PSNR↑ | SSIM↑ | PSNR↑ | SSIM↑ | PSNR↑ | SSIM↑ | PSNR↑ | SSIM↑ |
| DRUNet | 31.73 | 0.8865 | 31.48 | 0.8781 | 29.91 | 0.8412 | 28.60 | 0.8042 |
| SwinIR | 31.32 | 0.8856 | 31.05 | 0.8745 | 29.40 | 0.8326 | 28.07 | 0.7933 |
| Restormer | 31.06 | 0.8705 | 30.73 | 0.8612 | 29.38 | 0.8314 | 28.22 | 0.8001 |
| CODE | 31.65 | 0.8372 | 31.36 | 0.8276 | 29.75 | 0.7902 | 28.43 | 0.7543 |
| baseline | 32.96 | 0.8887 | 32.79 | 0.8851 | 31.97 | 0.8651 | 31.15 | 0.8419 |
| MT | 30.74 | 0.8176 | 30.70 | 0.8168 | 30.54 | 0.8133 | 30.36 | 0.8085 |
| **Ours** | **33.23** | **0.8956** | **33.09** | **0.8935** | **32.34** | **0.8774** | **31.56** | **0.8575** |

Table 9: Quantitative comparison on the McMaster (Zhang et al., 2011) dataset. Our RNINet outperforms other state-of-the-art methods in most cases, achieving significantly better performance in handling mixture noises, which are more complicated and representative in real application environments.

| Speckle Noise | $\sigma^2 = 0.02$ | | $\sigma^2 = 0.024$ | | $\sigma^2 = 0.03$ | | $\sigma^2 = 0.04$ | |
|---|---|---|---|---|---|---|---|---|
| Method | PSNR↑ | SSIM↑ | PSNR↑ | SSIM↑ | PSNR↑ | SSIM↑ | PSNR↑ | SSIM↑ |
| DRUNet | 29.31 | 0.8204 | 27.97 | 0.7868 | 26.34 | 0.7446 | 24.35 | 0.6876 |
| SwinIR | 28.88 | 0.8099 | 27.54 | 0.7772 | 25.98 | 0.7363 | 24.07 | 0.6810 |
| Restormer | 29.15 | 0.8276 | 28.12 | 0.8010 | 26.84 | 0.7667 | 25.17 | 0.7200 |
| CODE | 29.36 | 0.8150 | 28.06 | 0.7844 | 26.49 | 0.7451 | 24.56 | 0.6901 |
| baseline | 30.43 | 0.8567 | 29.63 | 0.8326 | 28.55 | 0.8005 | 27.05 | 0.7538 |
| MT | 29.90 | 0.8752 | 29.57 | 0.8678 | 28.99 | 0.8511 | 27.94 | 0.8144 |
| **Ours** | **31.21** | **0.8796** | **30.36** | **0.8574** | **29.17** | **0.8233** | **27.54** | **0.7742** |
| Salt & Pepper | $d = 0.002$ | | $d = 0.004$ | | $d = 0.008$ | | $d = 0.012$ | |
| Method | PSNR↑ | SSIM↑ | PSNR↑ | SSIM↑ | PSNR↑ | SSIM↑ | PSNR↑ | SSIM↑ |
| DRUNet | 32.6 | 0.9005 | 30.63 | 0.8736 | 28.08 | 0.8222 | 26.39 | 0.7736 |
| SwinIR | 31.87 | 0.9001 | 29.84 | 0.8726 | 27.24 | 0.8186 | 25.5 | 0.7653 |
| Restormer | 33.42 | 0.9475 | 30.97 | 0.9105 | 27.79 | 0.8409 | 25.89 | 0.7789 |
| CODE | 33.04 | 0.9015 | 31.24 | 0.8776 | 28.42 | 0.8255 | 26.49 | 0.773 |
| baseline | 33.07 | 0.8970 | 32.48 | 0.8873 | 31.30 | 0.8647 | 30.15 | 0.8371 |
| MT | 31.16 | 0.8746 | 31.06 | 0.8733 | 30.80 | 0.8692 | 30.48 | 0.8623 |
| **Ours** | **33.93** | **0.9092** | **33.21** | **0.8984** | **31.88** | **0.8752** | **30.64** | **0.8484** |
| Poisson Noise | $\alpha = 1.5$ | | $\alpha = 2$ | | $\alpha = 2.5$ | | $\alpha = 3$ | |
| Method | PSNR↑ | SSIM↑ | PSNR↑ | SSIM↑ | PSNR↑ | SSIM↑ | PSNR↑ | SSIM↑ |
| DRUNet | 32.25 | 0.8891 | 27.44 | 0.7500 | 23.84 | 0.6098 | 21.30 | 0.4997 |
| SwinIR | 32.15 | 0.8855 | 27.14 | 0.7397 | 23.69 | 0.6046 | 21.28 | 0.4988 |
| Restormer | 32.19 | 0.8917 | 28.70 | 0.7983 | 25.67 | 0.6949 | 23.52 | 0.6168 |
| CODE | 32.34 | 0.8796 | 27.47 | 0.7409 | 24.03 | 0.6091 | 21.67 | 0.5056 |
| baseline | 32.12 | 0.9025 | 29.34 | 0.8203 | 26.70 | 0.7183 | 24.48 | 0.6255 |
| MT | 30.55 | 0.8842 | 29.55 | 0.8696 | 27.79 | 0.8053 | 25.86 | 0.7171 |
| **Ours** | **32.69** | **0.9132** | **30.00** | **0.8452** | **27.10** | **0.7387** | **24.77** | **0.6384** |
| Mixture Noise | Level 1 | | Level 2 | | Level 3 | | Level 4 | |
| Method | PSNR↑ | SSIM↑ | PSNR↑ | SSIM↑ | PSNR↑ | SSIM↑ | PSNR↑ | SSIM↑ |
| DRUNet | 31.41 | 0.8946 | 31.19 | 0.8885 | 29.73 | 0.8568 | 28.50 | 0.8235 |
| SwinIR | 31.00 | 0.8935 | 30.79 | 0.8864 | 29.29 | 0.8523 | 28.06 | 0.8177 |
| Restormer | 31.08 | 0.8941 | 30.83 | 0.8869 | 29.52 | 0.8594 | 28.40 | 0.8298 |
| CODE | 31.73 | 0.8902 | 31.45 | 0.8824 | 29.85 | 0.8460 | 28.53 | 0.8089 |
| baseline | 32.06 | 0.8997 | 31.88 | 0.8974 | 31.17 | 0.8804 | 30.41 | 0.8596 |
| MT | 30.50 | 0.8797 | 30.40 | 0.8797 | 30.16 | 0.8782 | 29.91 | 0.8754 |
| **Ours** | **32.62** | **0.9070** | **32.45** | **0.9058** | **31.73** | **0.8930** | **30.99** | **0.8766** |

Table 10: Quantitative comparison on the CBSD68 (Martin et al., 2001) dataset. Our RNINet outperforms other state-of-the-art methods in most cases, achieving significantly better performance in handling mixture noises, which are more complicated and representative in real application environments.

| Speckle Noise | $\sigma^2 = 0.02$ | | $\sigma^2 = 0.024$ | | $\sigma^2 = 0.03$ | | $\sigma^2 = 0.04$ | |
|---|---|---|---|---|---|---|---|---|
| Method | PSNR↑ | SSIM↑ | PSNR↑ | SSIM↑ | PSNR↑ | SSIM↑ | PSNR↑ | SSIM↑ |
| DRUNet | 29.90 | 0.8041 | 28.43 | 0.7611 | 26.65 | 0.7063 | 24.49 | 0.6356 |
| SwinIR | 29.39 | 0.7906 | 27.92 | 0.7480 | 26.22 | 0.6950 | 24.17 | 0.6268 |
| Restormer | 29.73 | 0.8125 | 28.67 | 0.7820 | 27.28 | 0.7377 | 25.54 | 0.6843 |
| CODE | 30.06 | 0.8032 | 28.60 | 0.7623 | 26.86 | 0.7093 | 24.76 | 0.6404 |
| baseline | 31.08 | 0.8451 | 30.20 | 0.8170 | 29.07 | 0.7780 | 27.41 | 0.7200 |
| MT | 30.65 | 0.8737 | 30.33 | 0.8665 | 29.74 | 0.8484 | 28.59 | 0.8027 |
| **Ours** | **31.85** | **0.8697** | **30.95** | **0.8437** | **29.69** | **0.8033** | **27.90** | **0.7425** |
| Salt & Pepper | $d = 0.002$ | | $d = 0.004$ | | $d = 0.008$ | | $d = 0.012$ | |
| Method | PSNR↑ | SSIM↑ | PSNR↑ | SSIM↑ | PSNR↑ | SSIM↑ | PSNR↑ | SSIM↑ |
| DRUNet | 33.11 | 0.8910 | 31.13 | 0.8626 | 28.57 | 0.8070 | 26.86 | 0.7550 |
| SwinIR | 32.26 | 0.8894 | 30.14 | 0.8591 | 27.50 | 0.7979 | 25.73 | 0.7395 |
| Restormer | 33.79 | 0.9352 | 31.02 | 0.8907 | 27.94 | 0.8152 | 26.16 | 0.7542 |
| CODE | 33.61 | 0.8943 | 31.80 | 0.8702 | 28.89 | 0.8115 | 26.88 | 0.7532 |
| baseline | 33.58 | 0.8837 | 33.04 | 0.8749 | 31.93 | 0.8541 | 30.84 | 0.828 |
| MT | 31.71 | 0.8719 | 31.63 | 0.8709 | 31.42 | 0.8675 | 31.15 | 0.8616 |
| **Ours** | **34.33** | **0.8978** | **33.70** | **0.8880** | **32.49** | **0.8664** | **31.28** | **0.8406** |
| Poisson Noise | $\alpha = 1.5$ | | $\alpha = 2$ | | $\alpha = 2.5$ | | $\alpha = 3$ | |
| Method | PSNR↑ | SSIM↑ | PSNR↑ | SSIM↑ | PSNR↑ | SSIM↑ | PSNR↑ | SSIM↑ |
| DRUNet | 32.95 | 0.8788 | 27.53 | 0.7002 | 23.71 | 0.5444 | 21.09 | 0.4301 |
| SwinIR | 32.77 | 0.8716 | 27.21 | 0.6884 | 23.58 | 0.5406 | 21.09 | 0.4314 |
| Restormer | 32.83 | 0.8805 | 29.12 | 0.7665 | 25.95 | 0.6448 | 23.85 | 0.5630 |
| CODE | 33.08 | 0.8727 | 27.61 | 0.6951 | 23.93 | 0.5463 | 21.48 | 0.4359 |
| baseline | 32.81 | 0.8952 | 29.72 | 0.7881 | 26.77 | 0.6645 | 24.40 | 0.5578 |
| MT | 31.21 | 0.8817 | 30.26 | 0.8636 | 28.29 | 0.7800 | 26.03 | 0.6678 |
| **Ours** | **33.34** | **0.9048** | **30.39** | **0.8178** | **27.14** | **0.6831** | **24.66** | **0.5685** |
| Mixture Noise | Level 1 | | Level 2 | | Level 3 | | Level 4 | |
| Method | PSNR↑ | SSIM↑ | PSNR↑ | SSIM↑ | PSNR↑ | SSIM↑ | PSNR↑ | SSIM↑ |
| DRUNet | 32.14 | 0.8877 | 31.87 | 0.8816 | 30.33 | 0.8459 | 29.06 | 0.8083 |
| SwinIR | 31.60 | 0.8855 | 31.33 | 0.8776 | 29.76 | 0.8396 | 28.49 | 0.8001 |
| Restormer | 31.67 | 0.8844 | 31.38 | 0.8769 | 30.01 | 0.8470 | 28.88 | 0.8144 |
| CODE | 32.45 | 0.8848 | 32.10 | 0.8753 | 30.38 | 0.8343 | 29.01 | 0.7930 |
| baseline | 32.77 | 0.8915 | 32.62 | 0.8896 | 31.84 | 0.8727 | 31.06 | 0.8501 |
| MT | 31.16 | 0.8776 | 31.09 | 0.8774 | 30.88 | 0.8761 | 30.67 | 0.8737 |
| **Ours** | **33.26** | **0.8990** | **33.09** | **0.8976** | **32.37** | **0.8851** | **31.65** | **0.8686** |

Table 11: Quantitative comparison on the Kodak24 (Franzen, 1999) dataset. Our RNINet outperforms other state-of-the-art methods in most cases, achieving significantly better performance in handling mixture noises, which are more complicated and representative in real application environments.

| Speckle Noise | $\sigma^2 = 0.02$ | | $\sigma^2 = 0.024$ | | $\sigma^2 = 0.03$ | | $\sigma^2 = 0.04$ | |
|---|---|---|---|---|---|---|---|---|
| Method | PSNR↑ | SSIM↑ | PSNR↑ | SSIM↑ | PSNR↑ | SSIM↑ | PSNR↑ | SSIM↑ |
| DRUNet | 27.98 | 0.8036 | 26.63 | 0.7723 | 25.05 | 0.7339 | 23.16 | 0.6837 |
| SwinIR | 27.51 | 0.7931 | 26.19 | 0.7627 | 24.68 | 0.7256 | 22.88 | 0.6772 |
| Restormer | 28.21 | 0.8099 | 27.17 | 0.7851 | 25.86 | 0.7529 | 24.17 | 0.7106 |
| CODE | 28.06 | 0.7965 | 26.76 | 0.7674 | 25.26 | 0.7306 | 23.45 | 0.6825 |
| baseline | 29.52 | 0.8475 | 28.61 | 0.8227 | 27.51 | 0.7905 | 26.01 | 0.7464 |
| MT | 28.60 | 0.8831 | 28.25 | 0.8706 | 27.65 | 0.8466 | 26.64 | 0.8043 |
| **Ours** | **30.16** | **0.8696** | **29.25** | **0.8451** | **28.06** | **0.8113** | **26.48** | **0.7649** |
| Salt & Pepper | $d = 0.002$ | | $d = 0.004$ | | $d = 0.008$ | | $d = 0.012$ | |
| Method | PSNR↑ | SSIM↑ | PSNR↑ | SSIM↑ | PSNR↑ | SSIM↑ | PSNR↑ | SSIM↑ |
| DRUNet | 32.40 | 0.9199 | 30.47 | 0.8966 | 27.93 | 0.8509 | 26.25 | 0.8080 |
| SwinIR | 31.49 | 0.9171 | 29.43 | 0.8913 | 26.85 | 0.8405 | 25.16 | 0.7924 |
| Restormer | 32.80 | 0.938 | 30.25 | 0.9054 | 27.34 | 0.8481 | 25.63 | 0.7994 |
| CODE | 32.88 | 0.9154 | 31.08 | 0.8941 | 28.35 | 0.8484 | 26.49 | 0.8037 |
| baseline | 33.08 | 0.9137 | 32.35 | 0.9029 | 31.01 | 0.8791 | 29.80 | 0.8522 |
| MT | 29.75 | 0.8953 | 29.64 | 0.8932 | 29.38 | 0.8875 | 29.09 | 0.8802 |
| **Ours** | **33.48** | **0.9215** | **32.70** | **0.9102** | **31.32** | **0.8864** | **30.08** | **0.8606** |
| Poisson Noise | $\alpha = 1.5$ | | $\alpha = 2$ | | $\alpha = 2.5$ | | $\alpha = 3$ | |
| Method | PSNR↑ | SSIM↑ | PSNR↑ | SSIM↑ | PSNR↑ | SSIM↑ | PSNR↑ | SSIM↑ |
| DRUNet | 31.97 | 0.8872 | 26.93 | 0.7554 | 23.44 | 0.6442 | 21.02 | 0.5580 |
| SwinIR | 31.85 | 0.8815 | 26.60 | 0.7455 | 23.27 | 0.6390 | 20.96 | 0.5574 |
| Restormer | 31.94 | 0.8860 | 28.39 | 0.7970 | 25.33 | 0.7047 | 22.89 | 0.6269 |
| CODE | 31.98 | 0.8729 | 27.01 | 0.7452 | 23.69 | 0.6416 | 21.42 | 0.5619 |
| baseline | 31.91 | 0.9098 | 28.90 | 0.8231 | 26.33 | 0.7355 | 24.23 | 0.6600 |
| MT | 29.23 | 0.8961 | 28.39 | 0.8768 | 26.95 | 0.8144 | 25.33 | 0.7365 |
| **Ours** | **32.18** | **0.9218** | **29.50** | **0.8484** | **26.76** | **0.7537** | **24.57** | **0.6728** |
| Mixture Noise | Level 1 | | Level 2 | | Level 3 | | Level 4 | |
| Method | PSNR↑ | SSIM↑ | PSNR↑ | SSIM↑ | PSNR↑ | SSIM↑ | PSNR↑ | SSIM↑ |
| DRUNet | 31.36 | 0.9091 | 31.02 | 0.8977 | 29.39 | 0.8600 | 28.04 | 0.8239 |
| SwinIR | 30.85 | 0.9035 | 30.51 | 0.8909 | 28.84 | 0.8508 | 27.50 | 0.8141 |
| Restormer | 30.91 | 0.8953 | 30.62 | 0.8865 | 29.20 | 0.8562 | 28.07 | 0.8275 |
| CODE | 31.61 | 0.8961 | 31.19 | 0.8830 | 29.47 | 0.8432 | 28.08 | 0.8067 |
| baseline | 31.91 | 0.9120 | 31.67 | 0.9075 | 30.76 | 0.8852 | 29.89 | 0.8610 |
| MT | 29.23 | 0.8947 | 29.14 | 0.8942 | 28.91 | 0.8914 | 28.68 | 0.8874 |
| **Ours** | **32.11** | **0.9169** | **31.90** | **0.9146** | **31.10** | **0.8995** | **30.30** | **0.8799** |

Table 12: Quantitative comparison on the Urban100 (Huang et al., 2015) dataset. Our RNINet outperforms other state-of-the-art methods in most cases, achieving significantly better performance in handling mixture noises, which are more complicated and representative in real application environments.

