# OpenReview forum: "Random Is All You Need: Random Noise Injection on Feature Statistics for Generalizable Deep Image Denoising"
_ICLR.cc/2025/Conference — ICLR 2025 Poster_

### Official Review · Reviewer_YmqT · 2024-10-30

**Soundness:** 3
**Presentation:** 3
**Contribution:** 3
**Rating:** 6
**Confidence:** 4

**Summary:**

This paper introduces RNINet, an architecture for generalizable deep image denoising. The key innovation is the noise injection technique, which injects random noise into feature statistics and alters them to represent potential unseen noise domains. This allows the model to generalize well despite being trained only on Gaussian noise. The authors demonstrate RNINet's performance across multiple noise types and levels compared to both specialized and generalizable denoising methods. They also provide an analysis of the feature statistics to validate their approach.

**Strengths:**

+ The paper introduces a noise injection technique that directly manipulates feature statistics, which is an approach to improving generalization in image denoising.
+ This work surpasses previous methods, such as masked training denoising, on both in-distribution and some out-of-distribution datasets.
+ The authors used Deep Degradation Representation (DDR) for further analysis to evaluate the network's generalization capabilities.

**Weaknesses:**

- PSNR and SSIM have limitations in their accuracy of assessment in some aspects; the authors might consider adding more metrics, such as LPIPS.
- The paper could benefit from including more ablation studies to further explore the proposed method.

**Questions:**

- Regarding the experimental setup, the authors could try to train the model on a more general training set, rather than fixed Gaussian noise, and then compare the generalization ability.

---

> ### Author Response · Authors · 2024-11-18
> **Response to Reviewer YmqT**
>
> Thank you so much for your insightful comments and the recognition of our approach. We greatly appreciate the time and effort you've invested in reviewing our paper. We are eager to address the concerns raised by the reviewer as outlined below and are open to further discussions to resolve any remaining issues. *The issues are renumbered as 1, 2, 3, 4, etc., according to their original order presented by the reviewer.*
>
> **A1 $\rightarrow$ W1 & W2**: Thank you for your suggestion. The additional ablation study is included in Appendix A.2, where we adjusted the intensity of the random noise injected into feature statistics by applying a scaling factor. We plan to include LPIPS in the tables in the altered version.
>
> **A2 $\rightarrow$ Q1**: Regarding this key question raised by the reviewer, we recently conducted preliminary exploratory experiments to address this aspect. Specifically, we trained our RNINet on Gaussian noise first, followed by training on Poisson noise, and refer to this version as RNINet-g2p. The total number of iterations for RNINet-g2p was kept identical to the iterations used for training RNINet exclusively on Gaussian noise (referred to as RNINet-original). While we observed a very few cases with improvements, the overall results showed a significant performance drop, as presented below:
>
> | Speckle Noise  | ;$\sigma^2= 0.02 $ | $\sigma^2 = 0.024$ | $\sigma^2 = 0.03$ | $\sigma^2 = 0.04$ |
> |----------------|----------------|----------------|----------------|----------------|
> | Method         | PSNR↑/SSIM↑    | PSNR↑/SSIM↑    | PSNR↑/SSIM↑    | PSNR↑/SSIM↑    |
> | RNINet-g2p | 31.42/0.8425 | 30.39/0.8153 | 29.09/0.7801 | 27.48/0.7341 |
> | RNINet-original | 32.20/0.8674 | 31.29/0.8443| 30.04/0.8097 | 28.35/0.7639 |
>
> | Salt & Pepper | $d = 0.002 $ | $d = 0.004$    | $d = 0.008$| $d = 0.012 $ |
> |----------------|----------------|----------------|----------------|----------------|
> | Method         | PSNR↑/SSIM↑    | PSNR↑/SSIM↑    | PSNR↑/SSIM↑    | PSNR↑/SSIM↑    |
> | RNINet-g2p | 34.54/0.9290 | 32.93/0.9042 | 30.65/0.8554 | 29.00/0.8075 |
> | RNINet-original | 33.91/0.8992|33.04/0.8824|31.50/0.8486|30.11/0.8123 |
>
>
> | Poisson Noise  | $\alpha=1.5 $  | $\alpha=2$ | $\alpha=2.5$ | $\alpha=3$  |
> |----------------|----------------|----------------|----------------|----------------|
> | Method         | PSNR↑/SSIM↑    | PSNR↑/SSIM↑    | PSNR↑/SSIM↑  | PSNR↑/SSIM↑    |
> | RNINet-g2p | 32.72/0.8747 | 29.36/0.7658 | 26.80/0.6730 | 24.81/0.5970 |
> | RNINet-original | 33.65/0.9067|31.01/0.8337|28.19/0.7457|25.87/0.6673 |
>
> | Mixture Noise | $\text{Level 1}$ | $\text{Level 2}$| $\text{Level 3}$| $\text{Level 4}$|
> |----------------|----------------|----------------|----------------|----------------|
> | Method         | PSNR↑/SSIM↑    | PSNR↑/SSIM↑    | PSNR↑/SSIM↑    | PSNR↑/SSIM↑    |
> | RNINet-g2p | 32.79/0.8960 | 32.51/0.8880 | 31.24/0.8551 | 30.22/0.8227 |
> | RNINet-original | 33.23/0.8956|33.09/0.8935|32.34/0.8774|31.56/0.8575|
>
> |  | $\text{Synthetic ISP Noise}$ &emsp;| &emsp;| $\text{Monte Carlo Rendering Noise}$ ||
> |----------------|----------------|----------------|:----------------:|----------------|
> | Method         | PSNR↑/SSIM↑   &emsp; | PSNR↑/SSIM↑ (256 pp)| PSNR↑/SSIM↑  (128 pp)| PSNR↑/SSIM↑ (64 pp)   |
> | RNINet-g2p | 30.88/0.7963&emsp; | 31.74/0.8240|29.18/0.7477|26.47/0.6668|
> | RNINet-original | 31.15/0.8195&emsp;|33.36/0.8760| 30.34/0.8056| 27.38/0.7244
>
>
> In conclusion, we believe that training solely on Gaussian noise could obtain overall satisfactory results. However, there is considerable potential for further exploration into the optimal combination of training noise types to achieve the best results. This remains an interesting and open topic that we are actively considering. We are optimistic that future research will address this phenomenon in greater depth, both theoretically and experimentally.
>
> We hope we have adequately addressed all the reviewer's concerns through this rebuttal. Should there be any issues that remain unaddressed, please let us know, we look forward to further discussions with you.
>
> Best regards,
>
> Authors of #1857

---

> > ### Comment · Reviewer_YmqT · 2024-11-26
> >
> > Dear Author,
> >
> > Thank you for your detailed rebuttal and the additional experimental results. I found the new experiments quite interesting, and they provide further evidence to support your claims. These results enhance the overall persuasiveness of the article, and I would have been inclined to give a score of 7
> >
> > However, given the scoring system only allows discrete choices between 6 and 8, I have decided to maintain a score of 6 at this stage. My decision is primarily based on the expectation that the final version of the manuscript will include **a thorough discussion of the supplementary experiments, along with insights and reflections on the findings**. This would help ensure that the additional results are fully integrated into the narrative of the paper, further strengthening its overall contribution.
> >
> > Best regards,
> > Reviewer YmqT

---

> > > ### Author Response · Authors · 2024-11-27
> > > **Response to Reviewer YmqT**
> > >
> > > Dear Reviewer YmqT,
> > >
> > > Thank you very much for your thoughtful feedback and for your willingness to raise the score to 7 at this stage (though we understand that the system constraints only allow discrete choices between 6 and 8, and thus it cannot be reflected in the system now).
> > >
> > > In the next version of our manuscript (to be uploaded later), we will incorporate the supplementary experiments into the appendix first. Additionally, to align with your expectations, we will include a dedicated subsection in the appendix in the final version to provide a comprehensive discussion of the findings, supported by detailed textual descriptions and accompanying tables. We are continually working on refining this section to ensure it meets the highest standards.
> > >
> > > We sincerely appreciate your recognition and support of our work, and we hope to continue earning your stronger support in the next stage (if applicable).
> > >
> > > Thank you once again for your valuable input.
> > >
> > > Best regards,
> > >
> > > Authors of #1857

---

### Official Review · Reviewer_kYmE · 2024-11-01

**Soundness:** 2
**Presentation:** 3
**Contribution:** 2
**Rating:** 5
**Confidence:** 3

**Summary:**

The paper introduces RNINet, an architecture designed to improve generalization in deep image denoising. Unlike traditional denoising methods that often overfit specific noise types, RNINet incorporates a noise injection block to inject random noise into feature statistics during training, enabling the model to adapt to unseen noise types. RNINet enhances both denoising performance and computational efficiency, surpassing the Masked Training (MT) method. Extensive experiments demonstrate RNINet's superiority in handling various noise conditions while maintaining lower computational cost and achieving faster inference speeds.

**Strengths:**

- The paper presents a denoising model, RNINet, which achieves higher denoising performance than MT.
- RNINet demonstrates strong performance across diverse synthetic out-of-distribution (OOD) noise settings, highlighting its robustness beyond standard training conditions.
- RNINet demonstrates improved efficiency, operating at 0.1x the runtime of MT, an essential advantage as image denoising often precedes various downstream computer vision tasks.
- The paper is well-written and accessible, offering clear explanations and comprehensive ablation studies that enhance understanding.
- Authors provided the model to reproduce the results.

**Weaknesses:**

- The approach of injecting random noise into feature maps for enhanced generalization has been explored previously, and the addition of feature statistics such as mean and variance feels incremental rather than novel.
- If the primary objective is generalizable denoising, the model’s practicality in real-world scenarios should be further substantiated. Real-world evaluation is limited to the SIDD dataset, where performance metrics are relatively poor (although superior to the other methods in the table). Additionally, Figure 8 displays only a single, easy-to-denoise image that lacks sufficient visual details, making it unconvincing as evidence of real-world noise removal capabilities.
- While the paper emphasizes RNINet’s superiority over MT, there is insufficient theoretical explanation regarding how RNINet addresses MT’s limitations.
- The visual outcomes in Figure 5 do not appear substantially improved compared to MT, raising questions about the perceptual gains claimed. To enhance the clarity and persuasiveness of visual comparisons, could each visual example include quantitative metrics (e.g. PSNR, SSIM)?

**Questions:**

Please refer to Weaknesses section for major questions.

- Given that generalization to real-world conditions is the ultimate goal of the work, could the authors provide additional quantitative and qualitative results on various real-world benchmarks, such as the DND, Poly, CC datasets?

---

> ### Author Response · Authors · 2024-11-18
> **Response to Reviewer kYmE PART 1**
>
> Thank you so much for your insightful comments and for recognizing the superior performance of our approach. We greatly appreciate the time and effort you've invested in reviewing our paper. We are eager to address the concerns raised by the reviewer as outlined below and are open to further discussions to resolve any remaining issues. *The issues are renumbered as 1, 2, 3, 4, etc., according to their original order presented by the reviewer.*
>
> **A1 → W1**: Thank you for your comments. We would like to clarify that, to the best of our knowledge, our method is the first to introduce feature-level noise injection to enhance the generalizable performance of image denoising. Compared with previous methods, our method consistently outperforms them in both denoising performance and efficiency, with significant improvements. It is worth noting that, with the rapid evolution of research, discovering entirely unseen basic elements has become extremely hard. Consequently, the novelty in research should be increasingly recognized for its well-articulated motivation and innovative perspective. For instance, even renowned works like GANs, VAEs, and their applications often share similar basic components. However, this does not lessen the novelty of these studies due to their superior performance and the fresh perspectives they bring to their target tasks. We sincerely hope the reviewer will consider the novelty aspect of our work based on these points, which we believe align with the standards of most existing publications.
>
> **A2 → W2 & Q1**: Thank you for recognizing that our method outperforms other approaches in the SIDD evaluation. The zero-shot denoising is a highly challenging task on generalization capability and different from pipelines trained and tested on the same dataset (e.g., training on SIDD and testing on SIDD), these models can achieve exceptionally high quantitative metrics on specific datasets but they are overfitting them. For example, while the SIDD-pretrained Restormer model achieves around 40 dB PSNR on the SIDD testset, its performance drops significantly when simply tested on Speckle Noise, as shown in the table below:
>
> | Speckle Noise  | &emsp;$\sigma^2= 0.02 $ | &nbsp;&ensp;$\sigma^2 = 0.024$| &nbsp;&ensp;$\sigma^2 = 0.03$ | &nbsp;&ensp;$\sigma^2 = 0.04$ |
> |----------------|----------------|----------------|----------------|----------------|
> | Method  | PSNR↑/SSIM↑| PSNR↑/SSIM↑| PSNR↑/SSIM↑| PSNR↑/SSIM↑    |
> | Restormer (SIDD-pretrained) | 28.31/0.7913 | 27.62/0.7710 | 26.75/0.7460 | 25.62/0.7141 |
> | MT | 30.48/0.8094 | 30.20/0.7985 | 29.69/0.7776 | **28.66**/0.7360 |
> | Ours | **32.20/0.8674** | **31.29/0.8443** | **30.04/0.8097** | 28.35/**0.7639** |
>
> These results show that the performance of the SIDD-pretrained Restormer is also relatively poor compared to its 40 dB PSNR on the SIDD testset. Therefore, the outcomes heavily depend on the specific scenarios and *how we treat the results in a fair and consistent manner*. In the zero-shot generalization scenario (where model has no prior exposure to SIDD), we think our performance is satisfactory, as it surpasses previous methods. We are optimistic that future methods will continue to improve upon these results and achieve even greater performance.
>
> For additional experimental results on other real-world benchmarks, we would like to align with the suggestion from Reviewer Xna9, who brought LAN [1] to our attention. In LAN [1], two zero-shot denoisers are introduced and evaluated on two real-world datasets, Poly and CC (Nam), our work also targets this scenario. Following the experimental setup outlined in LAN [1], we compare our performance with the two zero-shot denoisers and the previous SOTA method, MT. We think this comparison provides a more convincing evaluation. The results are as below:
>
> | Datasets |  &emsp; &emsp; PolyU [2] | | &emsp; &emsp; CC (Nam) [3]  |  |
> |---------|-------------------|--------------------|--------------------|--------------------|
> | Methods | PSNR↑ | SSIM↑  | PSNR↑  | SSIM↑ |
> | ZS-Denoiser (ZS-N2N) [1] | 31.47| 0.8750 | 34.47 | 0.9020 |
> | ZS-Denoiser (Nbr2Nbr) [1] | 32.93 | 0.9110 | 35.39 | 0.9230 |
> | MT| 33.96| 0.9260| 33.20 | 0.9209 |
> | **Ours**| **37.59**| **0.9551**| **37.15**| **0.9525**|
>
> We believe the results above provide additional evidence of our method's strong generalization capability in handling zero-shot real-world denoising conditions. The DND dataset doesn't provide ground truth data but use online service for testing (we met a lot of troubles and request limitations), we currently have obtained results that: *RNINet (34.85/0.8391)* and *MT (33.75/0.8310)*, with our method outperforming the previous SOTA, MT, by a significant PSNR margin (*+1.1db*).
>
> [1] LAN: Learning to Adapt Noise for Image Denoising. CVPR 2024.
>
> [2] Real-world noisy image denoising: A new benchmark. ArXiv 2018.
>
> [3] A holistic approach to cross-channel image noise modeling and its application to image denoising. CVPR 2016.

---

> ### Author Response · Authors · 2024-11-18
> **Response to Reviewer kYmE PART 2**
>
> **A3 → W3**: Our work does not aim to address MT's limitations from a purely theoretical perspective. Instead, we tackle these limitations through a three-step approach: observation, model design with explanation, and experimentation.
>
> 1. **Observation**: We observed that MT tends to oversmooth image content, a side-effect that results in the loss of high-frequency details and a drop in PSNR (L52–L53 and Fig. 1).
>
> 2. **Model Design and Explanation**: We proposed RNINet, a framework fundamentally different from masked training. In Appendix A.3, we provide a theoretical explanation demonstrating the rationale and effectiveness of our design, proving that the key components of our approach are well-founded.
>
> 3. **Experimentation**: We conducted thorough experiments to address MT's limitations. The significant performance improvements achieved by our method substantiate its success in overcoming the challenges posed by MT.
>
> **A4 → W4 & W2 (partially)**: Thank you for your notification and suggestion. We find that in Fig. 5, some differences require zooming in to be noticeable. Considering the significant PSNR/SSIM improvements achieved by our RNINet,  including metrics like PSNR directly below each image is a very good suggestion. This would enhance the clarity and persuasiveness of the visual comparisons. We plan to apply this improvement to both Fig. 5 and Fig. 8, while also including more cases in Fig. 8. We are actively working on these updates and will reflect them in the final version of our manuscript.
>
> We hope we have adequately addressed all the reviewer's concerns through this rebuttal. Should there be any issues that remain unaddressed, please let us know, we look forward to further discussions with you.
>
> Best regards,
>
> Authors of #1857

---

> ### Author Response · Authors · 2024-11-19
> **Response to Reviewer kYmE: Updates in the New Version for Images and Tables**
>
> Dear Reviewer kYmE,
>
> Greetings!
>
> We would like to inform you that we have updated several figures and tables in our new version based on your suggestions. For example, we included more showcases in Figure 8 and added PSNR values below the images in Figure 5 to enhance the clarity and persuasiveness of the visual comparisons.
>
> Please refer to Appendix A.5 for details. You will find new figures and tables highlighted with "Supplementary for Rebuttals." Kindly review them for further information. Together with the previous rebuttal contents, we hope we have adequately addressed all of your concerns. Should there be any remaining issues, please let us know. We look forward to further discussions with you.
>
> Best regards,
>
> Authors of #1857

---

> ### Author Response · Authors · 2024-11-25
> **Response to Reviewer kYmE: Looking Forward to Your Feedback**
>
> Dear Reviewer kYmE,
>
> Thank you very much for your valuable suggestions. We have carefully followed your recommendations and incorporated the corresponding changes in both our rebuttal and the revised manuscript. We hope your concerns have been thoroughly addressed.
>
> As you may be aware, the discussion period is nearing its deadline, leaving us with very limited time for further exchanges. We would greatly appreciate it if you could take a moment to review our responses. If you find our updates satisfactory, we kindly hope you might consider updating your score.
>
> Should you have any additional concerns or require further clarification, we are more than happy to address them promptly.
>
> Thank you once again for your thoughtful feedback and support.
>
> Best regards,
>
> Authors of #1857

---

> > ### Comment · Reviewer_kYmE · 2024-11-25
> >
> > Dear Authors,
> >
> > Thank you for addressing my concerns in your detailed rebuttal and for updating the figures based on my suggestions. Your additional experiments have certainly strengthened the proposal of your paper. In particular, the zero-shot performance on the Poly and Nam datasets is commendable, showcasing significant improvements compared to existing methods.
> >
> > That said, my concerns regarding the novelty of the approach and its practical applicability remain unresolved. Therefore, I have decided to maintain my initial score of 5. While I find your arguments somewhat persuasive, I will adjust my confidence level from 4 to 3 to reflect this partial shift in my perspective.
> >
> > Best regards,
> > Reviewer kYmE

---

> > > ### Author Response · Authors · 2024-11-25
> > > **Response to Reviewer kYmE**
> > >
> > > Dear Reviewer kYmE,
> > >
> > > Thank you very much for your swift reply.
> > >
> > > We sincerely appreciate your recognition of our additional experiments. We are pleased that the zero-shot performance on additional real-world datasets has been acknowledged as commendable, showcasing significant improvements over existing methods. Given our superior zero-shot performance in handling real-world conditions, we believe this strongly supports the practical applicability of our approach, which was recently questioned by the reviewer.
> > >
> > > Regarding novelty, as we have clarified previously, **to the best of our knowledge, our method is the first to introduce feature-level noise injection to enhance the generalization performance of image denoising**. While we regret that we may not have been able to change your perspective on this point, we greatly value your efforts in helping us improve our manuscript.
> > >
> > > Once again, thank you for your thoughtful feedback and continued attention to our work.
> > >
> > > Best regards,
> > >
> > > Authors of #1857

---

### Official Review · Reviewer_Xna9 · 2024-11-02

**Soundness:** 2
**Presentation:** 2
**Contribution:** 2
**Rating:** 6
**Confidence:** 4

**Summary:**

- This paper enhances the generation performance of denoising by injecting random noise into the feature space, resulting in performance improvements over exisiting SOTA methods.

**Strengths:**

- Performance is improved simply by adding noise in the feature space.
- Builds upon existing experiments, effectively demonstrating the impact of this approach.

**Weaknesses:**

- The techincal contribution is quite incremental.
- Although noise injection at the feature level is effective, it is not particularly novel.
- For example. while Appendix A.3 theorectically demonstrates the effect of this approach, applying random noise to input data across various training datasets might yield similar results. In this regard, it may be challeging to assert that this method is definitively more effective than previous SOTA methods.
- While noise injection at the feature level in encoder can induce more nonlinearity in the noise distribution compared to input-only injection, claming this as a major technical strength might be an overstatement.
- This method seems applicable to most encoder-decoder architectures, which might be a strength over proposing a completely new architecture.

**Questions:**

- Aside from the noise injection module, are there other technical contributios that could be highlighted as strengths of the model?
- How does the model perform when noise is injected only into the image space (input)?
- Real-wolrd noise adaptation (e.g., LAN[1*]) is already in use. How does the model generalize to real-world noise? For instance, how does a model trained on SIDD peform on PolyU[2*] or NAM[3*] dataset?
- I consider the techinical contribution to be limited, but the performance is sufficiently high, so I gave it a rating of 5. I would consider increasing the rating if it can be shown to generalize well to real-world noise.
---
[1*] Kim, Changjin, Tae Hyun Kim, and Sungyong Baik. "LAN: Learning to Adapt Noise for Image Denoising." Proceedings of the IEEE/CVF Conference on Computer Vision and Pattern Recognition. 2024.
[2*] Jun Xu, Hui Li, Zhetong Liang, David Zhang, and Lei
Zhang. Real-world noisy image denoising: A new benchmark. arXiv preprint arXiv:1804.02603, 2018
[3*] Seonghyeon Nam, Youngbae Hwang, Yasuyuki Matsushita,
and Seon Joo Kim. A holistic approach to cross-channel image noise modeling and its application to image denoising.
In CVPR, 2016.

---

> ### Author Response · Authors · 2024-11-18
> **Response to Reviewer Xna9 PART 1**
>
> Thank you so much for your insightful comments and for recognizing the superior performance of our approach. We greatly appreciate the time and effort you've invested in reviewing our paper. We are eager to address the concerns raised by the reviewer as outlined below and are open to further discussions to resolve any remaining issues. *The issues are renumbered as 1, 2, 3, 4, etc., according to their original order presented by the reviewer.*
>
>
> **A1 → W1 & W2**: Thank you for your comments. We would like to clarify that, to the best of our knowledge, our method is the first to introduce feature-level noise injection to enhance the generalizable performance of image denoising. Compared with previous methods, our method consistently outperforms them in both denoising performance and efficiency, with significant improvements. It is worth noting that, with the rapid evolution of research, discovering entirely unseen basic elements has become extremely hard. Consequently, the novelty in research should be increasingly recognized for its well-articulated motivation and innovative perspective. For instance, even renowned works like GANs, VAEs, and their applications often share similar basic components. However, this does not lessen the novelty of these studies due to their superior performance and the fresh perspectives they bring to their target tasks. We sincerely hope the reviewer will consider the novelty aspect of our work based on these points, which we believe align with the standards of most existing publications.
>
> **A2 → W3 & Q2**: We guess there might be some misunderstandings regarding the random noise used in our experiments. In line with the task's requirements in previous works, we trained exclusively on Gaussian noise while testing on various unseen noise types. The random noise tensors injected into the feature statistics were also sampled from a normal/Gaussian distribution (L253-L254). For a fair comparison, all noise used in this task should be restricted to Gaussian noise. Therefore, adding random noise to the input data across training datasets (W3), or injecting noise into the image space (Q2), both equate to adding Gaussian noise to the input image, which is the standard pipeline for training a general Gaussian denoiser. However, our experiments have shown that a general Gaussian denoiser does not generalize as effectively as our approach. We include Appendix A.3 to enhance the interpretability of our method's effect. It does not solely demonstrate the effectiveness of our approach, but it does so in conjunction with the validation provided by our experiments. Thank you again for your notification, we would like to clarify these points to avoid any misunderstandings in our final version.
>
> **A3$\rightarrow$W4**: Thank you for this notification. To align with A1, we would like to revise this statement and emphasize our technical contributions as offering a fresh perspective by introducing noise injection for enhanced denoising generalization capability.
>
> **A4 → W5 & Q1**:  While not explicitly mentioned in our manuscript, we have tested our framework using a transformer-based encoder-decoder previously, which however resulted in a average performance drop of approximately 0.5dB and less satisfactory efficiency. We realize that the large model capacity with overtraining might cause overfitting to the training noise [a][b]. Therefore, we opted for a streamlined encoder-decoder framework as depicted in Fig. 3. The experiments demonstrate that the streamlined encoder-decoder associated with noise injection module contributes collectively to the superior performance in terms of both effectiveness and efficiency for our model.
>
> [a] Evaluating the generalization ability of super-resolution networks. TPAMI 2023.
>
> [b] Masked image training for generalizable deep image denoising. CVPR 2023.

---

> ### Author Response · Authors · 2024-11-18
> **Response to Reviewer Xna9 PART 2**
>
> **A5 → Q3 & Q4**: Thank you so much for bringing this brilliant work of LAN [1] to our attention. Upon detailed review of LAN [1], we found that it aims to bridge the noise distribution gap across various real-world denoising datasets to enhance generalization capabilities. Given that these datasets may have closely related noise distributions, LAN [1] also explores zero-shot denoising in Section 4.3 where the training and test noise distributions significantly differ. Our work also targets this scenario, thus we included comparisons with two zero-shot denoisers from LAN [1] alongside the previous SOTA method MT in our evaluations, we follow the experimental setup in LAN [1] to process PolyU [2] and Nam [3] testset, the results are presented as below:
>
> | Datasets |  &emsp; &emsp; PolyU [2] | | &emsp; &emsp; Nam [3]  |  |
> |---------|-------------------|--------------------|--------------------|--------------------|
> | Methods | PSNR↑ | SSIM↑  | PSNR↑  | SSIM↑ |
> | ZS-Denoiser (ZS-N2N) [1]    | 31.47  | 0.8750 | 34.47 | 0.9020 |
> | ZS-Denoiser (Nbr2Nbr) [1] | 32.93  | 0.9110 | 35.39  | 0.9230 |
> | MT      | 33.96   | 0.9260 | 33.20  | 0.9209 |
> | **Ours**    | **37.59** | **0.9551** | **37.15**  | **0.9525** |
>
> Our method demonstrated superior zero-shot generalization performance compared to MT and two zero-shot denoisers, which we believe provides strong evidence that our approach can generalize well to real-world noise. We would like to cite LAN [1] and other related benchmarks [2][3] and include the new evaluation results in our final version to further enhance the credibility of our findings.
>
> [1] LAN: Learning to Adapt Noise for Image Denoising. CVPR 2024.
>
> [2] Real-world noisy image denoising: A new benchmark. ArXiv 2018.
>
> [3] A holistic approach to cross-channel image noise modeling and its application to image denoising. CVPR 2016.
>
>
> **Supplement to Q1 & Q2**: We disable the noise injection on feature statistics and instead inject the noise into the image space, which is equivalent to training a general Gaussian denoiser as described in A2. Below, we present the test results on Speckle Noise:
>
> | Speckle Noise  | &emsp;$\sigma^2= 0.02 $      | &nbsp;&ensp;$\sigma^2 = 0.024$     | &nbsp;&ensp;$\sigma^2 = 0.03$      | &nbsp;&ensp;$\sigma^2 = 0.04$      |
> |----------------|----------------|----------------|----------------|----------------|
> | Method         | PSNR↑/SSIM↑    | PSNR↑/SSIM↑    | PSNR↑/SSIM↑    | PSNR↑/SSIM↑    |
> | RNINet (Image-Space) | 31.52/0.8485 | 30.64/0.8241 | 29.45/0.7902 | 27.92/0.7483 |
> | RNINet (Feature-Space) | **32.20/0.8674** | **31.29/0.8443** | **30.04/0.8097** | **28.35/0.7639** |
>
> The results show that RNINet with feature-space noise injection outperforms RNINet with image-space injection. This is because RNINet with image-space injection essentially degrades into a Gaussian denoiser. However, it still achieves reasonable results on Speckle Noise, which we believe demonstrates the effectiveness of the streamlined encoder-decoder framework in mitigating overfitting, thereby also contributing to the overall superior performance of our RNINet.
>
>
> We hope we have adequately addressed all the reviewer's concerns through this rebuttal. Should there be any issues that remain unaddressed, please let us know, we look forward to further discussions with you.
>
> Best regards,
>
> Authors of #1857

---

> > ### Comment · Reviewer_Xna9 · 2024-11-18
> >
> > Thank you for your detailed response. I am particularly impressed by the remarkable results under the same experimental settings as LAN. Highlighting this aspect would make the paper even stronger if revised. Therefore, I have adjusted my rating to a 6.

---

> > > ### Author Response · Authors · 2024-11-19
> > > **Response to Reviewer Xna9**
> > >
> > > Dear Reviewer Xna9,
> > >
> > > Thank you so much for your prompt response and for the score improvement!
> > >
> > > As the revision and preparation of the new version require careful checking, we are still working on it. In the next version (to be uploaded later), we will include the tables in the appendix first. Additionally, we would like to reorganize the content and include the experiments with LAN in the main experiments of our final version to highlight this aspect.
> > >
> > > Once again, thank you for your thoughtful insights and for recognizing the value of our work.
> > >
> > > Best regards,
> > >
> > > Authors of #1857

---

### Official Review · Reviewer_41dF · 2024-11-03

**Soundness:** 3
**Presentation:** 3
**Contribution:** 3
**Rating:** 6
**Confidence:** 3

**Summary:**

This paper revisits the challenge of generalizable image denoising, focusing on a training process that utilizes only Gaussian noise while testing encompasses various noise types. The authors reveal that models trained on different noise distributions yield distinct feature distributions. To enhance generalization capabilities, they introduce a novel neural network architecture that effectively handles multiple noise distributions when trained on white Gaussian noise. Central to this innovation is a noise injection block that integrates random noise into the features processed through the network layers.

Specifically, the architecture employs a U-Net type encoder-decoder structure, incorporating down and up convolutions, normalizations, ReLUs, and skip connections between the encoder and decoder. The noise injection blocks are strategically placed after each downsample bundle, which consists of Downsampling Convolution, Normalization, and ReLU.

In their experiments, the authors train the proposed network on white Gaussian noise with a sigma of 15, demonstrating its ability to generalize to various non-Gaussian noises, including (1) Speckle noise, (2) Salt & Pepper noise, (3) Poisson noise, (4) a mixture of noises (1)-(3) at different levels, (5) Image Signal Processing Noise, and (6) Monte Carlo rendered image noise.

**Strengths:**

- **Innovative Approach**: The paper offers a fresh perspective on the problem of image denoising.

- **Thorough Experimental Analysis**: The experimental analysis is extensive and well-structured.

- **Generalization Capability**: The proposed architecture effectively generalizes across various noise distributions, even when trained exclusively on white Gaussian noise. In the experiments conducted, it consistently outperforms competing methods in denoising performance. I believe that the concept of noise injection to enhance the generalization capabilities of image-denoising architectures holds significant potential and could attract interest from the research community.

**Weaknesses:**

I believe the most critical aspect of this task is its generalization capability. While the authors included the SIDD dataset in their experiments, they did not incorporate additional real-world noise datasets.

**Questions:**

In real-world scenarios, noise is highly spatially correlated and dependent on signal intensity, which inevitably incorporates image information, making it challenging to eliminate. From a theoretical standpoint, can Gaussian denoisers truly address the complexities of real noise?

---

> ### Author Response · Authors · 2024-11-18
> **Response to Reviewer 41dF**
>
> Thank you so much for your thoughtful comments and recognition of the innovative perspective and superior performance of our work. We greatly appreciate the time and effort you've invested in reviewing our paper. We are eager to address the concerns raised by the reviewer as outlined below and are open to further discussions to resolve any remaining issues.
>
> **A1 $\rightarrow$ W1**: Thank you for your suggestion. We have conducted experiments on two additional real-world noise datasets, PolyU [2] and Nam [3], aligned with the suggestion from Reviewer Xna9, who brought LAN [1] to our attention. In LAN [1], two zero-shot denoisers were introduced to evaluate performance under zero-shot real-world denoising conditions. Following the experimental setup outlined in LAN [1], we compare the performance of these two zero-shot denoisers with the previous state-of-the-art method, MT, and our proposed method. The results are presented below:
>
> | Datasets |  &emsp; &emsp; PolyU [2] | | &emsp; &emsp; Nam [3]  |  |
> |---------|-------------------|--------------------|--------------------|--------------------|
> | Methods | PSNR↑ | SSIM↑  | PSNR↑  | SSIM↑ |
> | ZS-Denoiser (ZS-N2N) [1] | 31.47| 0.8750 | 34.47 | 0.9020 |
> | ZS-Denoiser (Nbr2Nbr) [1] | 32.93 | 0.9110 | 35.39 | 0.9230 |
> | MT      | 33.96| 0.9260| 33.20 | 0.9209 |
> | **Ours**| **37.59**| **0.9551**| **37.15**| **0.9525**|
>
> We believe these results provide additional evidence of our method's strong generalization capability in handling zero-shot real-world denoising conditions.
>
> [1] LAN: Learning to Adapt Noise for Image Denoising. CVPR 2024.
>
> [2] Real-world noisy image denoising: A new benchmark. ArXiv 2018.
>
> [3] A holistic approach to cross-channel image noise modeling and its application to image denoising. CVPR 2016.
>
>
> **A2 $\rightarrow$ Q1**: For this question, we would like to provide our perspectives from two steps (Q&A).
>
> ***1.*** *Why do current renowned methods (SwinIR, Restormer, DRUNet, etc.) tend to conduct extensive evaluations on Gaussian noise rather than other types of noise?*
>
> Interestingly, many widely used models provide pretrained versions specifically for Gaussian noise, accompanied by extensive experiments. In our view, this preference can be attributed to two potential reasons:
>
> i) In many natural and technical processes, noise can often be regarded as the accumulation of multiple small effects. According to the Central Limit Theorem, when the sum of independent random variables becomes sufficiently large, their distribution converges to a normal (Gaussian) distribution.
>
> ii) Gaussian noise is characterized by a mean of 0 and a fixed variance, with a uniform spectrum across all frequencies. This means Gaussian noise can randomly and uniformly affect all inputs to a model, making it more robust to small input variations and enhancing its generalization capability. In other words, Gaussian noise is inherently more challenging to remove compared to other noise types, which may explain its frequent use in evaluations.
>
> ***2.*** *Can Gaussian denoisers address the complexities of real noise?*
>
> Real-world noise encompasses a diverse range of conditions and complexities, which are not fully captured by datasets like SIDD, PolyU, or Nam, as these primarily focus on camera sensor noise. In practice, other types of noise, such as Speckle noise, Poisson noise, and Gaussian noise, also occur in real-world scenarios. *General Gaussian denoisers are optimal for Gaussian noise but can address the complexities of real noise only moderately well, particularly in a zero-shot generalization context.* However, adjusted and generalizable Gaussian denoisers (*i.e.* RNINet, MT) can offer improved zero-shot solutions due to their robust generalization capabilities, making them better equipped to handle various forms of complex real-world noise. We are optimistic that there will be future research to thoroughly explore this intriguing question through both theoretical analysis and experimental validation.
>
> We hope we have adequately addressed all the reviewer's concerns through this rebuttal. Should there be any issues that remain unaddressed, please let us know, we look forward to further discussions with you.
>
> Best regards,
>
> Authors of #1857

---

> ### Author Response · Authors · 2024-11-28
> **Response to Reviewer 41dF: Looking Forward to Your Update**
>
> Dear Reviewer 41dF,
>
> Thank you for your valuable suggestions, recognition, and continued support of our work. In response to your comments, we have conducted additional experiments on two real-world datasets, PolyU and Nam, and compared them with two zero-shot denoisers in LAN. These experiments demonstrate the superior performance of our methods compared to previous approaches under very practical conditions. Furthermore, we have elaborated on your question regarding the use of Gaussian denoisers for real noise. For additional insights on this issue, you may also refer to the findings in our rebuttal to Reviewer YmqT, who found them quite interesting. We hope our rebuttal effectively addresses your concerns and enhances our original explanations.
>
> As you may be aware, the discussion period after extension is nearing its deadline, leaving us with limited time for further exchanges. Your feedback is immensely valuable to us, and we are eager to hear your thoughts. If you find our updates satisfactory, sincerely, we wonder if you could consider updating/raising your score/confidence to support our work?
>
> Should you require any further clarification or additional information, please don’t hesitate to reach out.
>
> Thank you again for your support and consideration.
>
> Best regards,
>
> Authors of #1857

---

### Author Response · Authors · 2024-11-18
**General Response to All Reviewers**

Dear Reviewers,

Greetings!

We sincerely appreciate the efforts and valuable feedback provided by all four reviewers, which have significantly contributed to improving the quality and clarity of our manuscript. We are pleased that the reviewers recognized the key strengths of our work, including our innovative introduction of noise injection for generalizable image denoising, offering a fresh perspective on this problem (**41dF, YmqT**); the good readability of the paper, as well as the thoroughness and generality of the experiments and analysis (**41dF, kYmE, YmqT**); and the superior performance of our method compared to previous state-of-the-art methods (**All Reviewers: 41dF, Xna9, kYmE, YmqT**).

In response to the reviewers’ suggestions, we have incorporated additional results and expanded our experiments to further demonstrate the effectiveness of our proposed method. Here, we would like to provide a brief summary and clarify that:

1. To the best of our knowledge, our method is the first to introduce feature-level noise injection to enhance the generalizable performance of image denoising. Consistent with the settings in previous methods, our approach is trained exclusively on Gaussian noise yet effectively generalizes across various noise distributions. It demonstrates strong performance across diverse out-of-distribution (OOD) noise settings, consistently outperforming competing methods in both denoising performance and efficiency.

2. A common request from the reviewers (41dF, Xna9, kYmE) was to include evaluations on additional real-world noise datasets. We appreciate Reviewer Xna9 for bringing LAN [1] to our attention. LAN [1] aims to bridge the noise distribution gap across various real-world denoising datasets to enhance generalization capabilities, and it introduces and discusses two zero-shot denoisers. In response, we included comparisons with the two zero-shot denoisers from LAN [1] alongside the previous state-of-the-art method, MT, in our evaluations. We follow the experimental setup in LAN [1] to process PolyU [2] and Nam (CC) [3] testset, and the results are presented as below:

| Datasets |  &emsp; &emsp; PolyU [2] | | &emsp; &emsp; Nam (CC) [3]  |  |
|---------|-------------------|--------------------|--------------------|--------------------|
| Methods | PSNR↑ | SSIM↑  | PSNR↑  | SSIM↑ |
| ZS-Denoiser (ZS-N2N) [1] | 31.47| 0.8750 | 34.47 | 0.9020 |
| ZS-Denoiser (Nbr2Nbr) [1] | 32.93 | 0.9110 | 35.39 | 0.9230 |
| MT      | 33.96| 0.9260| 33.20 | 0.9209 |
| **Ours**| **37.59**| **0.9551**| **37.15**| **0.9525**|

These results are also available in our response to the corresponding reviewers. We believe these findings provide additional evidence of our method's strong generalization capability in addressing zero-shot real-world denoising conditions.

[1] LAN: Learning to Adapt Noise for Image Denoising. CVPR 2024.

[2] Real-world noisy image denoising: A new benchmark. ArXiv 2018.

[3] A holistic approach to cross-channel image noise modeling and its application to image denoising. CVPR 2016.

3. Minor issues. We also take valuable suggestions from the reviewers, including: adding metrics like PSNR directly below each image to enhance the clarity and persuasiveness of visual comparisons (kYmE), revising specific statements for clarity to avoid misunderstandings (Xna9), etc. We are continually working on these improvements to ensure the best possible writing and presentation of our manuscript. Additionally, we will recheck these minor issues to ensure they are thoroughly addressed in the final version.

For detailed rebuttals, please refer to our responses to each reviewer.

Once again, thank you for your thoughtful insights and continued attention. We look forward to further discussions with you!

Best regards,

Authors of #1857

---

### Author Response · Authors · 2024-11-25
**General Response: Looking Forward to Your Feedback**

Dear Reviewers,

Thank you sincerely for your time and effort in reviewing our work. As you may be aware, the discussion period this year concludes on November 26, leaving us with limited time to address any remaining concerns or questions.

We would greatly appreciate the opportunity to engage with you during this period. If you haven’t already done so, we kindly request you to review the content of our rebuttals. If our responses address your concerns satisfactorily, we hope you might consider updating your score accordingly.

Your feedback is immensely valuable to us, and we are eager to hear your thoughts. If you require further clarification or additional information, please don’t hesitate to reach out.

Thank you again for your support and consideration.

Best regards,

Authors of #1857

---

### Public Comment · ~Zhengwei_Yin1 · 2025-07-16

Code available at: https://github.com/default-8541/RNINet

---

### Meta-Review · Area_Chair_P92n · 2024-12-23

**Metareview:**

This paper introduces RNINet built on a streamlined encoder-decoder framework to enhance both efficiency and overall performance of generalizable deep image denoising. To do so, observing that feature statistics such as mean and variance shift in response to different noise conditions, the authors train a pure RNINet (only simple encoder-decoder) on individual noise types. Then,  Leveraging these insights, the authors incorporate a noise injection block that injects random noise into feature statistics within our framework, significantly improving generalization across unseen noise types.

This work has four reviewers. Three reviewers are positive to accept it, while the other reviewer is negative. After checking the responses, I find that most of the concerns raised by the reviewers are also well addressed. In this regard, this work can be accepted.

**Additional Comments On Reviewer Discussion:**

Four reviewers raised many concerns about the generalization capability on additional real-world noise datasets, incremental technical novelty, insufficient theoretical explanation, more metrics (such as LPIPS), more ablation studies, and so on.

After the rebuttal, most of the concerns of four reviewers are well addressed, and thus three reviewers are positive to accept it.

---

### Decision · Program_Chairs · 2025-01-22

Accept (Poster)